# Diverse interactions and ecosystem engineering can stabilize community assembly

Justin D. Yeakel [1,2✉], Mathias M. Pires [3], Marcus A. M. de Aguiar[3], James L. O'Donnell[4], Paulo R. Guimarães Jr.[5], Dominique Gravel[6] & Thilo Gross[7,8,9,10]

The complexity of an ecological community can be distilled into a network, where diverse interactions connect species in a web of dependencies. Species interact directly with each other and indirectly through environmental effects, however to our knowledge the role of these ecosystem engineers has not been considered in ecological network models. Here we explore the dynamics of ecosystem assembly, where species colonization and extinction depends on the constraints imposed by trophic, service, and engineering dependencies. We show that our assembly model reproduces many key features of ecological systems, such as the role of generalists during assembly, realistic maximum trophic levels, and increased nestedness with mutualistic interactions. We find that ecosystem engineering has large and nonlinear effects on extinction rates. While small numbers of engineers reduce stability by increasing primary extinctions, larger numbers of engineers increase stability by reducing primary extinctions and extinction cascade magnitude. Our results suggest that ecological engineers may enhance community diversity while increasing persistence by facilitating colonization and limiting competitive exclusion.

[1] University of California Merced, 5200 Lake Road, Merced, CA 95343, USA. [2] Santa Fe Institute, 1399 Hyde Park Road, Santa Fe, NM 87501, USA. [3] Universidade Estadual de Campinas, Cidade Universitária Zeferino Vaz-Barão Geraldo, Campinas, São Paulo 13083-970, Brazil. [4] University of Washington, Seattle, WA 98195, USA. [5] Universidade de São Paulo, Cidade Universitária, São Paulo-State of São Paulo, São Paulo, Brazil. [6] Universitè de Sherbrooke, 2500 Boulevard de l'Université, Sherbrooke, QC J1K 2R1, Canada. [7] University of California, Davis, CA 95616, USA. [8] Alfred-Wegener-Institut Helmholtz-Zentrum für Polar- und Meeresforschung, Oldenburg, Germany. [9] Helmholtz Institute for Functional Marine Biodiversity at the University of Oldenburg (HIFMB), Ammerländer Heerstrasse 231, 26129 Oldenburg, Germany. [10] University of Oldenburg, ICBM, 26129 Oldenburg, Germany. ✉email: jdyeakel@gmail.com

To unravel nature's secrets we must simplify its abundant complexities and idiosyncrasies. The layers of natural history giving rise to an ecological community can be distilled—among many forms—into a network, where nodes represent species and links represent interactions between them. Networks are generally constructed for one type of interaction, such as food webs capturing predation[1–3] or pollination networks capturing a specific mutualistic interaction[4], and continue to lead to significant breakthroughs in our understanding of the dynamical consequences of community structure[5–7]. This perspective has also been used to shed light on the generative processes driving the assembly of complex ecological communities[8,9].

To what extent assembly leaves its fingerprint on the structure and function of ecological communities is a source of considerable debate[10–12]. There is strong evidence that functional traits constrain assembly[12–14], while differences in species' trophic niche[15,16], coupled with early establishment of fast/slow energy channels[17], appear to significantly impact long-term community dynamics. There has been growing interest in understanding the combined role of trophic and mutualistic interactions in driving assembly[18,19], where the establishment of species from a source pool[19–21] and the plasticity of species interactions[22–25] constrain colonization and extinction dynamics. While recent interest in "multilayer networks" comprising multiple interaction types (multitype interactions) may provide additional insight into these processes[26,27], there is not yet a well-defined theory for the assembly of communities that incorporates multitype interactions, as well as both biotic and abiotic components from which functioning ecosystems are composed (cf. ref. [28]).

Diverse interactions occur not only between species but indirectly through the effects that species have on the abiotic environment[29–31]. Elephants root out large saplings and small trees, enabling the formation and maintenance of grasslands[32,33] and creating habitat for smaller vertebrates[34]. Burrowing rodents such as gophers and African mole rats create shelter and promote primary production by aerating the soil[35,36], salmon, and aquatic invertebrates create freshwater habitats by changing stream morphology[37], and leaf-cutter ants alter microclimates, influencing seedling survival and plant growth[38]. These examples illustrate ecosystem engineering, where the engineering organism alters the environment on timescales longer than its own[39]. Engineers are widely acknowledged to have impacts on both small and large spatial scales[40], and likely serve as important keystone species in many habitats[41].

Ecosystem engineering not only impacts communities on ecological timescales, but has profoundly shaped the evolution of life on Earth[42]. For example, the emergence of multicellular cyanobacteria fundamentally altered the atmosphere during the Great Oxidation Event of the Proterozoic roughly 2.5 Byrs BP[42,43], paving the way for the biological invasion of terrestrial habitats. In the oceans it is thought that ribosomal RNA (rRNA) and protein biogenesis of aquatic photoautotrophs drove the nitrogen:phosphorous ratio (the Redfield Ratio) to ca. 16:1 matching that of plankton[44], illustrating that engineering clades can have much larger, sometimes global-scale effects.

The effect of abiotic environmental conditions on species is commonly included in models of ecological dynamics[45–47] due to its acknowledged importance and because it can—to first approximation—be easily systematized. By comparison the way in which species engineer the environment defies easy systemization due to the multitude of mechanisms by which engineering occurs. While interactions between species and the abiotic environment have been conceptually described[30,48], the absence of engineered effects in network models was detailed by Odling-Smee et al.[31], where they outlined a conceptual framework that included both species and abiotic compartments as nodes of a network, with links denoting both biotic and abiotic interactions.

How does the assembly of species constrained by multitype interactions impact community structure and stability? How are these processes altered when the presence of engineers modifies species' dependencies within the community? Here, we model the assembly of an ecological network where nodes represent ecological entities, including engineering species, non-engineering species, and the effects of the former on the environment, which we call abiotic "modifiers." The links of the network that connect both species and modifiers represent trophic ("eat" interactions), service ("need" interactions), and engineering dependencies, respectively (Fig. 1; see "Methods" for a full description). Trophic interactions represent both predation and parasitism, whereas service interactions account for non-trophic interactions associated with reproductive facilitation such as pollination or seed dispersal. In our framework, a traditional mutualism (such as a plant-pollinator interaction) consists of a service (need) interaction in one direction and a trophic (eat) interaction in the other. These multitype interactions between species and modifiers thus embed multiple dependent ecological sub-systems into a single network (Fig. 1). Modifiers in our framework overlap conceptually with the "abiotic compartments" described in Odling-Smee et al.[31]. Following Pillai et al.[49], we do not track the abundances of biotic or abiotic entities but track only their presence or absence. We use this framework to explore the dynamics of ecosystem assembly, where the colonization and extinction of species within a community depends on the constraints imposed by the trophic, service, and engineering dependencies. We then show how observed network structures emerge from the process of assembly, compare their attributes with those of empirical systems, and examine the effects of ecosystem engineers.

Our results offer four key insights into the roles of multitype interactions and ecosystem engineering in driving community assembly. First, we show that the assembly of communities in the absence of engineering reproduces many features observed in empirical systems. These include changes in the proportion of generalists over the course of assembly that accord with measured data and trophic diversity similar to empirical observations. Second, we show that increasing the frequency of mutualistic interactions leads to the assembly of ecological networks that are more nested, a common feature of diverse mutualistic systems[50], but that are also prone to extinction cascades. Our third key result shows that increasing the proportion of ecosystem engineers within a community has nonlinear effects on observed extinction rates. While we find that a low amount of engineering increases extinction rates, a high amount of engineering has the opposite effect. Finally we show that redundancies in engineered effects promote community diversity by lowering the barriers to colonization.

## Results and discussion

**Assembly without ecosystem engineering**. Our framework assumes that communities assemble by random colonization from a source pool. A species from the source pool can colonize if it finds at least one resource that it can consume (one eat interaction is satisfied; cf. ref. [51]) and all of its non-trophic needs are met (all need interactions are satisfied; see Fig. 1). As such, service interactions are assumed to be obligate, whereas trophic interactions are flexible—except in the case of a consumer with just a single resource. While an abiotic basal resource is always assumed to be present (white node in Fig. 1b), following the establishment of an autotrophic base, the arrival of mixotrophs (i.e., mixing auto- and heterotrophy) and lower-trophic heterotrophs create opportunities for organisms occupying higher trophic levels to invade. This expanding niche space initially serves as an accelerator for community growth.

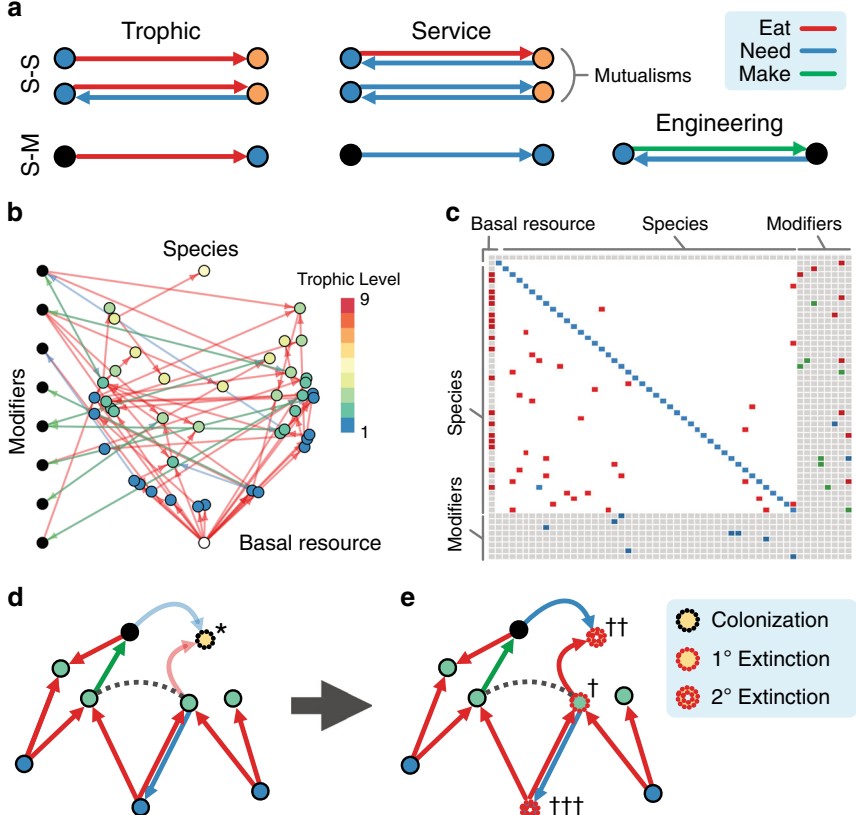

**Fig. 1 Model framework for ecological networks with multitype interactions and ecosystem engineering. a** Multitype interactions between species (colored nodes) and abiotic modifiers (black nodes). Trophic and mutualistic relationships define both species–species (S–S) and species–modifier (S–M) interactions; an engineering interaction is denoted by an engineer that makes a modifier, such that the modifier needs the engineer to persist. **b** An assembling food web with species (color denotes trophic level) and modifiers. The basal resource is the white node at the bottom of the network. **c** The corresponding adjacency matrix with colors denoting interactions between species and modifiers. **d** A species (*) can colonize a community when a single trophic and all service requirements are met. **e** Greater vulnerability increases the risk of primary extinction via competitive exclusion (competition denoted by dashed line) to species (†). The extinction of species (†) will cascade to affect those connected by trophic (††) and service (†††) dependencies.

Following the initial colonization phase, extinctions begin to slow the rate of community growth. Primary extinctions occur if a given species is not the strongest competitor for at least one of its resources. A species' competition strength is determined by its interactions: competition strength is enhanced by the number of need interactions (where the number of potential and realized interactions are equivalent) and penalized by the number of its realized resources (i.e., those resources present in the local community, favoring functional trophic specialists) and realized predators (i.e., those predators present in the local community). This encodes three key assumptions: that mutualisms provide a fitness benefit[52], specialists are stronger competitors than generalists[53–56], and having many predators entails an energetic cost[57]. Secondary extinctions occur when a species loses its last trophic or any of its service requirements. As the colonization and extinction rates converge, the community reaches a steady state around which it oscillates (Fig. 2a). See Fig. 1d, e for an illustration of the assembly process, and the "Methods" and Supplementary Note 1 for a complete description. Specific model parameterizations are described in Supplementary Note 2.

Assembly of ecological communities in the absence of engineering results in interaction networks with structures consistent with empirical observations. As the community reaches steady state (Fig. 2a), we find that the connectance of trophic interactions ($C(t) = L(t)/S(t)^2$, where $S(t)$ is species richness and $L(t)$ is the number of links at time $t$) decays to a

constant value (Supplementary Fig. 1). Decaying connectance followed by stabilization around a constant value has been documented in the assembly of mangrove communities[16] and experimental aquatic mesocosms[17]. The initial decay is likely inevitable in sparse webs as early in the assembly process the small set of tightly interacting species will have a high link density from which it will decline as the number of species increases. In Supplementary Note 3 we include a brief comparison of assembly model food webs with those produced by the Niche model[58]. While the aims of these approaches are quite distinct, we provide this comparison as a reference point to traditional food web models, and to emphasize that both approaches result in food webs with similar structures (Supplementary Figs. 2 and 3).

Recent empirical work has suggested that generalist species may dominate early in assembly, whereas specialists colonize after a diverse resource base has accumulated[16,51]. Here, the trophic generality of species $i$ is defined as $G_i(t) = k_i^{in}(t)/(L^*/S^*)$[58], where $k_i^{in}(t)$ is the number of resource species linked to consumer $i$ at simulation time-step $t$, which is scaled by the steady state link density $L^*/S^*$, as is typically performed in empirical investigations[16]. Only trophic links between species are considered here, such that we ignore links to the abiotic basal resource in our evaluation of trophic generality. A species is classified as a generalist if $G_i > 1$ and a specialist if $G_i < 1$. If generality is evaluated with respect to the steady state link density, we find that species with many potential trophic interactions realize only a

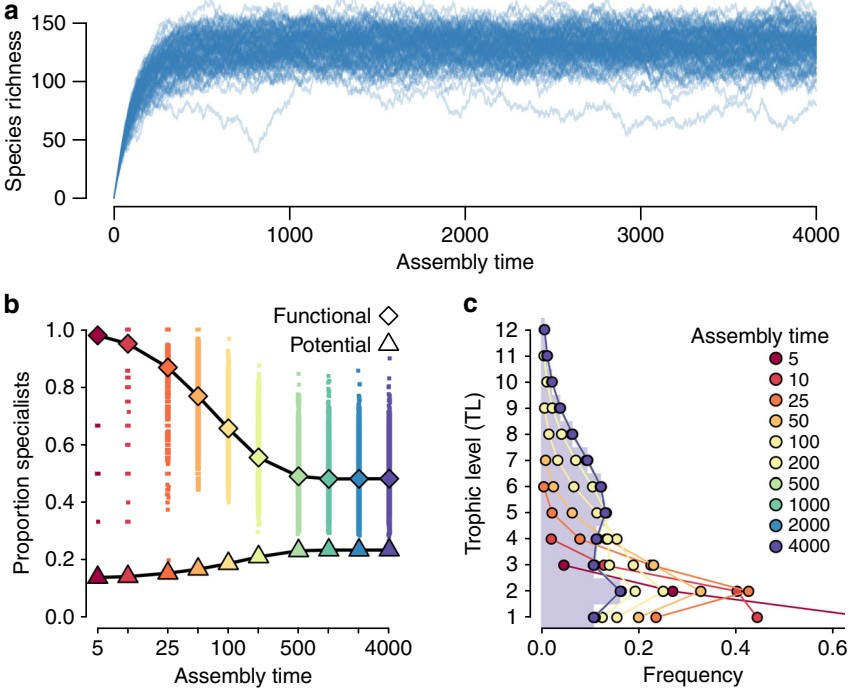

**Fig. 2 Food web structure over the course of assembly. a** Assembling communities over time from a pool of 200 non-engineering species. Steady state species richness is reached by $t = 250$. **b** The proportion of specialists as a function of assembly time (iterations). Diamonds denote expected values for functional (realized) trophic interactions at each point in time, and triangles denote expected values for potential trophic interactions (as if all trophic interactions with all species in the pool were realized), where the expectation is taken across replicates. Individual replicate results are shown for functional trophic interactions (small points). **c** The frequency distribution of trophic levels as a function of assembly time (iterations). Autotrophs occupy $TL = 1$. Measures were evaluated across $10^4$ replicates; see "Methods" for parameter values.

subset of them, thereby functioning as specialists early in the assembly process (Fig. 2b). As the community grows, more potential interactions become realized, and functional specialists become functional generalists. Moreover, as species assemble, the available niche space expands, and the proportion of potential trophic specialists grows (Fig. 2b). This latter observation confirms expectations from the trophic theory of island biogeography[51], where communities with lower richness (i.e., early assembly) are less likely to support specialist consumers than species-rich communities (late assembly). At steady state the proportion of functional specialists is ca. 48%, which is similar to empirical observations of assembling mangrove island food webs[16].

The dominance of functional specialists following the initial assembly of autotrophs is due to the colonization of lower-trophic consumers with few resources, where the observed trophic level (TL) distribution early in assembly ($t = 5$) has an average $TL = 1.6$[59]. Four trophic levels are typically established by $t = 50$, where colonization is still dominant, and by the time communities reach steady state the interaction networks are characterized by an average $TL_{max}$ ($\pm$standard deviation) $= 11 \pm 2.8$ (Fig. 2c). While the maximum trophic level is higher than that measured in most consumer-resource systems[60], it is not unreasonable if parasitic interactions (which we do not differentiate from other consumers) are included[61]. Overall, the most common trophic level among species at steady state is ca. $TL = 4.75$.

The distribution of trophic levels changes shape over the course of assembly. Early in assembly, we observe a skewed pyramidal structure, where most species feed from the base of the food web. At steady state, we observe that intermediate trophic levels dominate, with frequencies taking on an hourglass structure (purple bars, Fig. 2c). Compellingly, the trophic richness pyramids that we observe at steady state follow closely the

hourglass distribution observed for empirical food webs and are less top-heavy than those produced by static food web models[62].

**Structure and dynamics of mutualisms**. Nested interactions, where specialist interactions are subsets of generalist interactions, are a distinguishing feature of mutualistic networks[50,63–65]. Nestedness has been shown to maximize the structural stability of mutualistic networks[66], emerge naturally via adaptive foraging behaviors[24,67] and neutral processes[68], and promote the influence of indirect effects on coevolutionary dynamics[69]. While models and experiments of trophic networks suggest that compartmentalization confers greater stabilizing properties[70,71], interaction asymmetry among species may promote nestedness in both trophic[65] and mutualistic systems[72]. Processes that operate on different temporal and spatial scales may have a significant influence on these observations[73]. For example, over evolutionary time, coevolution and speciation may degrade nested structures in favor of modularity[25], and there is some evidence from Pleistocene food webs that geographic insularity may reinforce this process[74].

Does the assembly of ecological networks favor nestedness when mutualistic interactions are frequent? In the absence of mutualisms, the trade-offs in our model preclude high levels of nestedness because we assume that generalists are at a competitive disadvantage when they share the same resources with a specialist consumer. Yet, we find that as we increase the frequency of service interactions (holding constant trophic interaction frequency; see Supplementary Note 2), the assembled community at steady state becomes more nested (Fig. 3a). More service interactions increase a species' competition strength, lowering its primary extinction risk. Participation in a mutualism thus delivers a fitness advantage to the species receiving the service, compensating for the lower competitive strength of

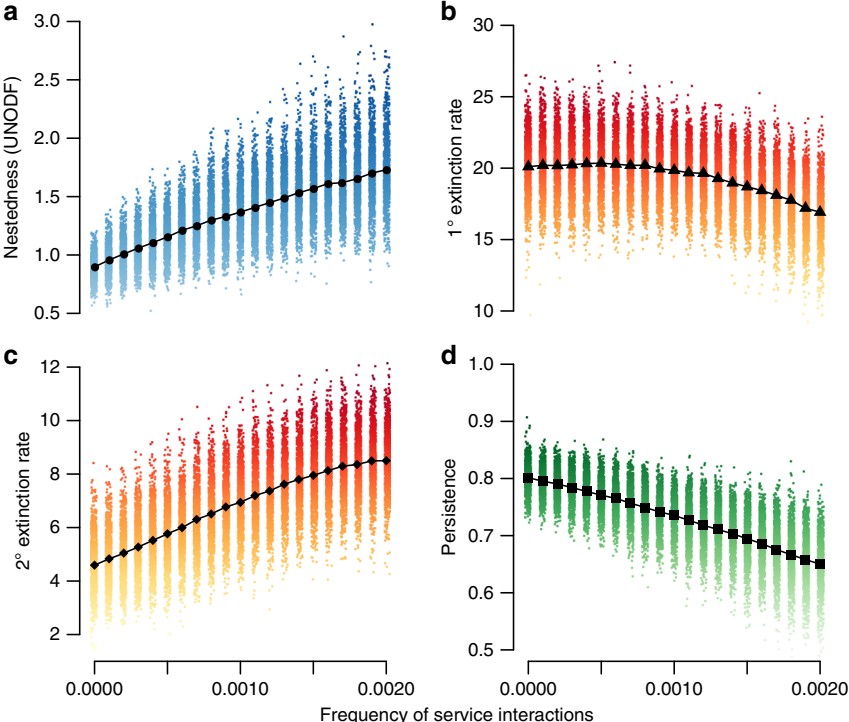

**Fig. 3 Community structure and stability as a function of the frequency of service interactions. a** Structural nestedness of communities, measured as UNODF (Unipartite Nestedness based on Overlap and Decreasing Fill; measured using the R package UNODF v.1.2)[100]. The value reported is the mean value taken across the rows and columns of the adjacency matrix accounting for both trophic and service interactions. **b** Mean rate of primary extinction (where primary extinctions occur from competitive exclusion of consumers over shared resources) and **c** secondary extinction (which cascade from primary extinctions) as a function of service interaction frequency. **d** Species persistence as a function of service interaction frequency. Primary and secondary extinction rates were evaluated at the community level, whereas persistence was determined for each species and averaged across the community. Measures were evaluated for $10^4$ replicates; see "Methods" and Supplementary Note 2 for parameter values.

generalists and allowing generalists to share subsets of resources with specialists, promoting nestedness. However, increases in mutualisms also increase inter-species dependencies, which raises the potential risk associated with losing mutualistic partners[75,76]. While this shifting landscape of extinction risks lowers the steady state species richness of highly mutualistic communities, we do not observe a direct relationship between nestedness and richness (Supplementary Fig. 4).

When we examine the dynamics of the community as a function of service interaction frequency, we observe that mutualistic interactions have different effects on primary versus secondary extinction rates. As service dependencies bolster the competitive strength of otherwise susceptible species such as trophic generalists and species with multiple predators, the rate of primary extinctions is lowered, though this effect is weak (Fig. 3b). However, because mutualisms build rigid dependencies between species, more service interactions result in higher frequencies of secondary extinctions (Fig. 3c). In communities with many mutualistic interactions, this combined influence yields extinctions that are less likely to occur, but that lead to larger cascades when they do.

An increased rate of secondary extinctions means that the network is less robust to perturbation, which may impact community turnover, or persistence. If we measure persistence in terms of the proportion of time species are established in the community, we find that higher frequencies of service interactions lower average persistence (increased species turnover; Fig. 3d). Analysis of species-specific interactions reveals that it is the species that require more services that have lower persistence (Supplementary Fig. 5). Some empirical systems appear to support model predictions. For example, long-term observations of ant-plant mutualistic systems have demonstrated

high rates of turnover among service-receivers (plants) relative to service-donors (ants)[77].

We emphasize that we have restricted ourselves to examining the effects of obligate mutualisms, although the importance of non-obligate mutualisms has long been recognized[23,24,67,78,79]. We expect that the increased rate of secondary extinctions attributable to the loss of obligate mutualistic partners to have greater impact on system stability than the potential loss of non-obligate mutualistic partners. As such, we do not expect inclusion of non-obligate mutualisms to alter the qualitative nature of our findings.

**Assembly with ecosystem engineering.** The concept of ecosystem engineering, or more generally niche construction, has both encouraged an extended evolutionary synthesis[80] while also garnering considerable controversy[81,82]. Models that explore the effects of ecosystem engineering are relatively few, but have covered important ground[31,39]. For example, engineering has been shown to promote invasion[83], alter primary productivity[84], and change the selective environment over eco-evolutionary timescales[85,86], which can lead to unexpected outcomes such as the fixation of deleterious alleles[87]. On smaller scales, microbiota construct shared metabolitic resources that have a significant influence on microbial communities[88], the dynamics of which may even serve as the missing ingredient stabilizing some complex ecological systems[89]. Soil is one place where these macro- and microbiotic systems intersect[90]. Many microbes and detriti-vores transform and deliver organic matter into the macrobiotic food web, themselves hosting a complex network of trophic and service dependencies between species and abiotic entities[91,92].

We next explore the effects of ecosystem engineering by allowing species to produce abiotic modifiers as additional nodes

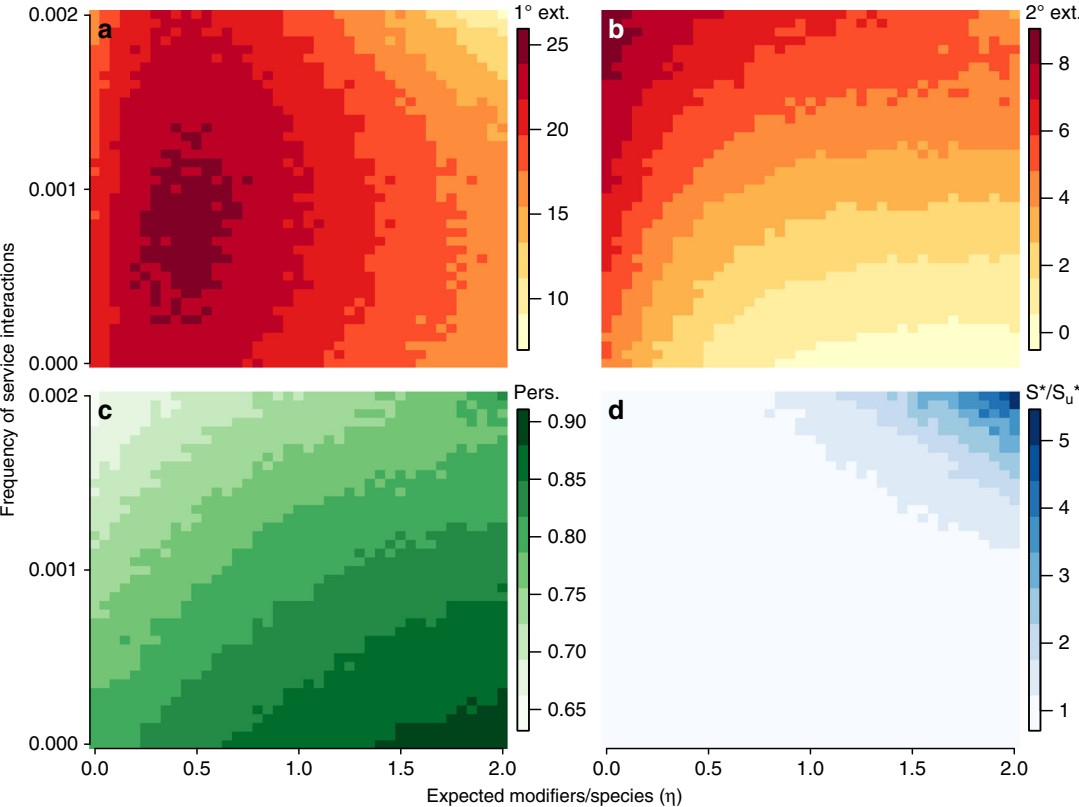

**Fig. 4 Community stability as a function of the frequency of service interactions and modifiers per species. a** Mean rates of primary extinction, where primary extinctions occur from competitive exclusion of consumers over shared resources. **b** Mean rates of secondary extinction, which cascade from primary extinctions. **c** Mean species persistence. **d** The ratio $S^*/S_u^*$, where $S_u^*$ denotes steady states for systems where all engineered modifiers are unique to each engineer, and $S^*$ denote steady states for systems with redundant engineering. Higher values of $S^*/S_u^*$ mean that systems with redundant engineers have higher richness at the steady state than those without redundancies. Primary and secondary extinction rates were evaluated at the community level, whereas persistence was determined for each species and averaged across the community. Each measure reports the expectation taken across 50 replicates. See "Methods" and Supplementary Note 2 for parameter values.

in the ecological network (Fig. 1). These modifier nodes produced by engineers can serve to fulfill resource or service requirements for other species. The parameter $\eta$ defines the mean number of modifiers produced per species in the pool, drawn from a Poisson distribution (see "Methods" and Supplementary Note 1 for details). If a species makes ≥1 modifier, we label it an engineer. As the mean number of modifiers/species $\eta$ increases, both the number of engineers in the pool, as well as the number of modifiers made per engineer increases. As detailed in Supplementary Note 1, multiple engineers can make the same modifier, such that engineering redundancies are introduced when $\eta$ is large. When an engineer colonizes the community, so do its modifiers, which other species in the system may interact with. When engineers are lost, their modifiers will also be lost, though can linger in the community for a period of time inversely proportional to the density of disconnected modifiers in the community (see Supplementary Note 1).

While the inclusion of engineering does not significantly impact the structure of species–species interactions within assembling food webs (see Supplementary Note 4 and Supplementary Fig. 6), it does have significant consequences for community stability. Importantly, these effects also are sensitive to the frequency of service interactions within the community, and we find that their combined influence can be complex.

As the number of engineers increases, mean rates of primary extinction are first elevated and then decline (Fig. 4a). At the same time, the mean rates of secondary extinction systematically decline and persistence systematically increases (Fig. 4b, c). When engineered modifiers are rare ($0 < \eta \le 0.5$), higher rates of primary extinction coupled with lower rates of secondary extinction mean that extinctions are common, but of limited magnitude such that disturbances are compartmentalized. As modifiers become more common both primary and secondary extinction rates decline, which corresponds to increased persistence. We suggest two mechanisms that may produce the observed results. First, when engineers and modifiers are present but rare, they provide additional resources for consumers. This stabilization of consumers ultimately results in increased vulnerability of prey, such that the cumulative effect is increased competitive exclusion of prey and higher rates of primary extinction (Fig. 4a). Second, when engineers and their modifiers are common ($\eta > 0.5$) the available niche space expands, lowering competitive overlap and suppressing both primary and secondary extinctions. Notably the presence of even a small number of engineers serves to limit the magnitude of secondary extinction cascades (Fig. 4b). Assessment of species persistence as a function of trophic in-degree (number of resources) and out-degree (number of consumers) generally supports this proposed dynamic (Supplementary Fig. 7).

Increasing the frequency of service interactions promotes service interactions between species and engineered modifiers (Fig. 1). A topical example of the latter is the habitat provided to invertebrates by the recently discovered rock-boring teredinid shipworm (*Lithoredo abatanica*)[93]. Here, freshwater invertebrates are serviced by the habitat modifications engineered by the shipworm, linking species indirectly via an abiotic effect (in our framework via a modifier node). As the frequency of service

interactions increases, the negative effects associated with rare engineers is diminished (Fig. 4a). Increasing service interactions both elevates the competitive strength of species receiving services (from species and/or modifiers), while creating more inter-dependencies between and among species. As trophic interactions are replaced by service interactions, previously vulnerable species gain a competitive foothold and persist, lowering rates of primary extinctions (Fig. 4a). The cost of these added services to the community is an increased rate of secondary extinctions (Fig. 4b) and higher species turnover (Fig. 4c), such that extinctions are less common but lead to larger cascades.

While the importance of engineering timescales has been emphasized previously[39], redundant engineering has been assumed to be unimportant[94]. We argue that redundancy may be an important component of highly engineered systems, and particularly relevant when the effects of engineers increase their own fitness[83] as is generally assumed to be the case with niche construction[86]. If ecosystem engineering also includes, for example, biogeochemical processes such as nitrogen-fixing among plants and mycorrhizal fungi, redundancy may be perceived as the rule rather than the exception. Moreover, the vast majority of contemporary ecosystem engineering case studies focus on single taxa, such that redundant engineers appear rare[94]. If we consider longer timescales, diversification of engineering clades may promote redundancy, and in some cases this may feed back to accelerate diversification[95]. Such positive feedback mechanisms likely facilitated the global changes induced by cyanobacteria in the Proterozoic[42,43] among other large-scale engineering events in the history of life[42]. Engineering redundancies are likely important on shorter timescales as well. For example, diverse sessile epifauna on shelled gravels in shallow marine environments are facilitated by the engineering of their ancestors, such that the engineered effects of the clade determine the future fitness of descendants[96]. In the microbiome, redundant engineering may be very common due to the influence of horizontal gene transfer in structuring metabolite production[97]. In these systems, redundancy in the production of shared metabolitic resources may play a key role in community structure and dynamics[88,89].

When there are few engineers, each modifier in the community tends to be unique to a particular engineering species. Engineering redundancies increase linearly with $\eta$ (Supplementary Note 1 and Supplementary Fig. 8), such that the loss of an engineer will not necessarily lead to the loss of engineered modifiers. We examine the effects of this redundancy by comparing our results to those produced by the same model, but where each modifier is uniquely produced by a single species. Surprisingly, the lack of engineering redundancies does not alter the general relationship between engineering and measures of community stability (Supplementary Fig. 9). However, we find that redundancies play a central role in maintaining species diversity. When engineering redundancies are allowed, steady state community richness $S^*$ does not vary considerably with increasing service interactions and engineering (Supplementary Fig. 10a). In contrast, when redundant engineering is not allowed (each modifier is unique to an engineer, denoted by the subscript "u"), steady state community richness $S_u^*$ declines sharply (Fig. 4d and Supplementary Fig. 10b).

Communities lacking redundant engineering have lower species richness because species' trophic and service dependencies are unlikely to be fulfilled within a given assemblage (Supplementary Fig. 10c, d). Colonization occurs only when trophic and service dependencies are fulfilled. A species requiring multiple engineered modifiers, each uniquely produced, means that each required entity must precede colonization. This magnifies the role of priority effects in constraining assembly order[12], precluding many species from colonizing. In contrast, redundant engineering increases the

temporal stability of species' niches while minimizing priority effects by allowing multiple engineers to fulfill the dependencies of a particular species. Our results thus suggest that redundant engineers may play important roles in assembling ecosystems by lowering the barriers to colonization, promoting community diversity.

## Conclusions

We have shown that simple process-based rules governing the assembly of species with multitype interactions can produce communities with realistic structures and dynamics. Moreover, the inclusion of ecosystem engineering by way of modifier nodes reveals that low levels of engineering may be expected to produce higher rates of extinction while limiting the size of extinction cascades, and that engineering redundancy—whether it is common or rare—serves to promote colonization and by extension community diversity. We suggest that including the effects of engineers, either explicitly as we have done here, or otherwise, is vital for understanding the inter-dependencies that define ecological systems. As past ecosystems have fundamentally altered the landscape on which contemporary communities interact, future ecosystems will be defined by the influence of engineering today. Given the rate and magnitude with which humans are currently engineering environments[98], understanding the role of ecosystem engineers is thus tantamount to understanding our own effects on the assembly of natural communities.

## Methods

**Assembly model framework**. We model an ecological system with a network where nodes represent "ecological entities" such as populations of species and or the presence of abiotic modifiers affecting species. Following Pilai et al.[49], we do not track the abundances of entities but track only their presence or absence (see also refs. [19,20]). The links of the network represent interactions between pairs of entities (x, y). We distinguish three types of such interactions: x eats y, x needs y to be present, x makes modifier y.

The assembly process entails two steps: first a source pool of species is created, followed by colonization/extinction into/from a local community. The model is initialized by creating $S$ species and $M = \eta S$ modifiers, such that $N = S + M$ is the expected total number of entities (before considering engineering redundancies) and $\eta$ is the expected number of modifiers made per species in the community, where the expectation is taken across independent replicates. For each pair of species (x, y) there is a probability $p_e$ that x eats y and probability $p_n$ that x needs y. For each pair of species x and modifier m, there is a probability $q_e$ that species x eats modifier m and a probability $q_n$ that species x needs modifier m. Throughout we assume that $p_e = q_e$ and $p_n = q_n$ for simplicity. Each species $i$ makes a number of modifiers $M_i \sim \text{Poiss}(\eta)$. If engineering redundancies are allowed, once the number of modifiers per species is determined each modifier is assigned to a species independently to match its assigned number of modifiers. This means that multiple species may make the same modifier, and that there may be some modifiers that are not assigned to any species, which are eliminated from the pool. Accounting for engineering redundancies, the number of modifiers in the pool becomes $M' = \eta S(e-1)/e$ where $e$ is Euler's number. If engineering redundancies are not allowed, each modifier is made by a single engineer and $M' = M$.

In addition to interactions with ecosystem entities, there can be interactions with a basal resource, which is always present. The first species always eats this resource, such that there is always a primary producer in the pool. Other species eat the basal resource with probability $p_e$. Species with zero assigned trophic interactions are assumed to be primary producers. See Supplementary Note 1 for additional details on defining the source pool.

We then consider the assembly of a community, which at any time will contain a subset of entities in the pool and always the basal resource. In time, the entities in the community are updated following a set of rules. A species from the pool can colonize the community if the following conditions are met: (1) all entities that a species needs are present in the community, and (2) at least one entity that a species eats is present in the community. If a colonization event is possible, it occurs stochastically in time with rate $r_c$.

An established species is at risk of extinction if it is not the strongest competitor at least one of its resources that it eats. We compute the competitive strength of species $i$ as

$$\sigma_i = c_n n_i - c_e e_i - c_v v_i, \qquad (1)$$

where $n_i$ is the number of entities that species $i$ needs, $e_i$ is the number of entities from the pool that species $i$ can eat, and $v_i$ is the number of species in the community that eat species $i$. This captures the ecological intuition that mutualisms

provide a fitness benefit[52], specialists are stronger competitors than generalists[55], and many predators entail an energetic cost[57]. The coefficients $c_n$, $c_e$, $c_v$ describe the relative effects of these contributions to competition strength. In the following, we use the relationship $c_n > c_e > c_v$, such that the competitive benefit of adding an additional mutualism is greater than the detriment incurred by adding another resource or predator. A species at risk of extinction leaves the community stochastically in time at rate $r_e$.

A modifier is present in the community whenever at least one species that makes the modifier is present. If a species that makes a modifier colonizes a community, the modifier is introduced as well; however; modifiers may persist for some time after the last species that makes the modifier goes extinct. Any modifier that has lost all of its makers disappears stochastically in time at rate $r_m$.

The model described here can be simulated efficiently with an event-driven simulation utilizing a Gillespie algorithm. In these types of simulations, one computes the rates $r_j$ of all possible events $j$ in a given step. One then selects the time at which the next event happens by drawing a random number from an exponential distribution with mean $1/\Sigma_j r_j$. At this time, an event occurs that is randomly selected from the set of possible events such that the probability of event $a$ is $r_a/\Sigma_j r_j$. The effect of the event is then realized and the list of possible events is updated for the next step. This algorithm is known to offer a much better approximation to the true stochastic continuous time process than a simulation in discrete time steps, while providing a much higher numerical efficiency[99]. Simulations described in the main text have default parameterizations of $S = 200$, $p_e = 0.01$, $c_n = \pi$, $c_e = \sqrt{2}$, $c_v = 1$, and 4000 iterations. Replicates are defined as the independent assembly of independently drawn source pools with a given parameterization.

**Reporting summary**. Further information on research design is available in the Nature Research Reporting Summary linked to this article.

## Data availability

Simulation data to reproduce the findings of this study can be generated from the code available for download at https://github.com/jdyeakel/Lego.

## Code availability

The custom simulation code supporting this work is available for download at https://github.com/jdyeakel/Lego.

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

## Acknowledgements

We would like to thank Uttam Bhat, Irina Birskis Barros, Emmet Brickowski, Jennifer A. Dunne, Ashkaan Fahimipour, Marilia P. Gaiarsa, Jean Philippe Gibert, Christopher P Kempes, Eric Libby, Lauren C. Ponisio, Taran Rallings, Samuel V. Scarpino, Megha Suswaram, and Ritwika VPS for insightful discussions and comments throughout the lengthy gestation of this manuscript. The original idea was conceived at the Networks on Networks Working Group in Göttingen, Germany (2014) and the Santa Fe Institute (2015). This work was formerly prepared as a part of the Ecological Network Dynamics Working Group at the National Institute for Mathematical and Biological Synthesis (2015–2019), sponsored by the National Science Foundation through NSF Award DBI-1300426, with additional support from The University of Tennessee, Knoxville. Infinite revisions were conducted at the Santa Fe Institute made possible by travel awards to J.D.Y. and T.G. Additional support came from UC Merced startup funds to J.D.Y., the International Center for Theoretical Physics ICTP-SAIFR, FAPESP (2016/01343-7 and 2019/20271-5) and CNPq (302049/2015-0) to M.A.M.d.A., CNPq and FAPESP (2018/14809-0) to P.R.G., and DFG research unit 1748 and EPSRC (EP/N034384/1) to T.G.

## Author contributions

J.D.Y. and T.G. conceived of the model framework. J.D.Y., M.M.P., M.A.M.d. A., and T.G. designed the analyses. J.D.Y., M.M.P., M.A.M.d.A., J.L.O.D., P.R.G., D.G., and T.G. analyzed the results and contributed to multiple versions of the manuscript.

## Competing interests

The authors declare no competing interests.
