## [Peer Review File · Nature Communications]

Reviewers' Comments:

Reviewer #1:

Remarks to the Author:

I think this manuscript is very interesting in its approach to understand how ecosystem engineering affects networks of interacting species. The inclusion of more non-trophic interactions into such networks is extremely important if we ever want to understand the complexity of real ecological communities. Overall the manuscript reads very well and provides some interesting insights. I am in general very excited about this piece of work, but I have some questions especially about the inclusion of engineering interactions.

Major comments:

Ecosystem engineering - as a very general description of a phenomenon where species modify their physical environment- has the benefit of making people/researchers well aware of it's importance but this concept includes a vast number of very different kind of interactions that operate at different scales. I wonder if the authors should at least spend a few sentences on that issue. For example, a tree that provides the structure for all species that live on it will lead to a massive extinction cascades if it disappears, but grazers that shorten the vegetation (e.g. rabbits) and thereby changing the soil temperature might have a less global influence. Maybe this could be addressed by changing the influence of single modifiers on different proportion of the species present? I apologise if this has been somehow included and I misunderstood.

I wasn't entirely sure how the engineering effect is included, as both positive and negative? As far as I understand this is here modelled as resources/conditions that are provided. But these can in effect impact species in very different ways. E.g. shading can allow some plant species to grow better but may slow down feeding in herbivore insects.

It would be interesting to see how ecosystem engineering affects network structure as well, does the model provide any ideas on that? Are there any structural features that might stabilize as found for other types of interactions?

I suspect that the estimations of secondary extinctions are very conservative as it ignores community dynamics driving secondary extinction cascades.

Minor comments

Abstract: please include why ecosystem engineering increases the likelihood of

extinction cascades.

Line 128 I guess trophic interactions are only flexible for generalists not specialists, maybe be more specific.

Line 141 why does increased number of prey reduce competitiveness? Is the assumption that a specialist is better adapted?

Line 344 This points to a negative impact of engineering on the initial set of species. But as far as I can see there are no negative effects of modifiers on species survival (e.g. this has been documented in a number of studies recoding the impact of invasive ecosystem engineer species). Does that make the model too simplistic?

Line 315 Does that mean more links from an engineer or more engineered species (links from modifier to species)?

Reviewer #2:

Remarks to the Author:

Referee report for "Diverse interactions and ecosystem engineering stabilize community assembly" by Yeakel et al.

This manuscript presents an ecological network assembly model, able to include feeding interactions, service interactions, and interactions mediated through ecosystem engineering. The authors analyze the model's performance, i.e. the realism of the generated network structure; they explore how structure and stability are affected by service interactions (mutualisms); and, finally, how engineering affects stability (primary and secondary extinctions; persistence; species richness at steady state).

There are four main claims. Firstly, the assembly model results in interaction networks with structures consistent with empirical observations. Secondly, increasing the frequency of mutualisms, in networks with feeding and service interactions, results in a more nested interaction structure at steady-state species richness and a lower average persistence of species in the network. Thirdly, ecosystem engineering tends to decrease primary and secondary extinctions (somewhat dependent on the frequency of engineering and service interactions). Fourthly, engineering redundancies (more than one species engineer the same entity) increases steady-state species richness through the facilitation of colonization.

To my knowledge, the main claims are novel, particularly with regards to the effects of ecosystem engineering, and are well supported by the presented data. The paper is timely and connects to the growing interest in non-trophic interactions and how they affect network structure and dynamics. The paper will certainly interest researchers in network ecology and is likely to influence thinking in the field.

For the reasons outlined above, I recommend the paper to be considered for publication in Nature communications. Due to its timeliness, novelty, and likelihood to garner interest, it is a very good fit for the journal. However, before publication, the authors need to address some issues. These issues relate more to presentation than to scientific soundness but are still vital to address (see below). I, therefore, recommend that the editor invites the authors to revise their manuscript to address these concerns before making a final decision.

Here, I briefly list the main issues (for details, see the attached full referee report):

- 1) The introduction is insufficient. Firstly, the introduction does not provide background for the first half of the study (Sections "Assembly without ecosystem engineering" and "Structure and dynamics of mutualisms"). Secondly, the authors never outline the research questions.
- 2) The method description is insufficient and opaque on several points. Furthermore, there are inconsistencies between the Methods section and Appendix 1 in describing the same aspects of the model.
- 3) There are, in some cases, inconsistencies between main text and appendix result graphs, as well as between text and result graphs.
- 4) Though the main claims are sufficiently supported, the suggestions of the underlying mechanisms are not. Some speculation into mechanism is fine, as long as the authors present it as such, but I do not find that this is the case here.

Again, I wish to emphasize that I consider these issues to be related to oversights or mistakes in the presentation, and not to faults in the science proper. As the manuscript merits further consideration, I will advise on the requested additional points:

The manuscript is in parts clearly written, in parts not. In the detailed referee comments, I have identified particular paragraphs and sentences that need revising. It is also possible that what I have mentioned as being a lack of support for the mechanisms driving the main results is simply a matter of unclear writing. Again, I have made comments on these sections, asking for clarified verbal reasoning or data to support them. I do not consider any further "experiments", i.e. simulations, to be necessary, but a deeper analysis of existing data is, in some cases, desired (see detailed review comments).

The manuscript cannot be shortened much. If anything, the clarifications I request may require it to grow somewhat in size. However, I have identified a few paragraphs that the authors might consider for removal or shortening:

- Lines 45-56; this paper is not directly concerned with the evolutionary importance of ecosystem engineers.
- The introductory paragraphs in some of the sections seem rather broad in scope. The authors could potentially streamline them by cutting out some of the content that is not directly relevant to the paper. Paragraphs to consider for such streamlining occur on lines 210-228, 293-308, and 380-406.

The authors have done themselves justice without overselling their main claims. As far as I can tell, they are fair in their treatment of previous literature and discuss their claims appropriately in the context of previous literature, with two exceptions. Firstly, there should be some background (and hence references) that motivate the exploration of the topics of the first two result sections (i.e. model results without engineers). Secondly, the authors do not discuss the behavior of the model in relation to other assembly models, which is odd, especially given the focus of the first result section. The authors should address these two gaps in order to meet the literature-related requirements.

The authors have not provided sufficient methodological detail for the experiments to be reproduced, but the omissions are not great and should be easy to amend. In the detailed referee comments, I ask the authors to provide more methodological information on specific aspects.

There are no statistical analyses of the data, only summary statistics. Statistical analyses are not strictly necessary here. However, the descriptions of the summary statistics should be improved (see detailed comments on figures).

There are no special ethical concerns.

Detailed referee comments for the manuscript “Diverse interactions and ecosystem engineering stabilize community assembly” by Yeakel et al.

There are a total of 50 comments. About half are minor comments or related only to the clarity of writing. The remaining comments, each marked by a bolded topic sentence, are mostly related to structural issues with the writing, to missing method information, to lacking support for result claims (in this case, specifically related to the suggested mechanisms driving the main results, not the main results *per se*), or to discrepancies in the result presentation.

Abstract

1. I do not agree that “ecosystem engineering...increases the magnitude of secondary extinction cascades.” What is driving the increased magnitude of secondary extinction cascades are the service interactions, not the engineering interactions (see Fig. 4b). The fact that some of these service interactions are between species and modifiers, rather than between species only, does not change the causality.

Section 1: “Introduction”

2. Rephrase lines 80-82: “...service interactions account for all non-trophic interactions such as pollination or seed dispersal.” All service interactions within this framework are positive for the receiver; service interactions lower the receiving species’ extinction risk. Therefore, the service interactions cannot “account for all non-trophic interactions” as non-trophic interactions include negative ones, e.g. predator interference and fear of predation.
3. **The introduction needs re-writing to better reflect what the study is about and to describe what the specific research questions are.** The introductory paragraphs, lines 1-70, lead the reader to expect the study to be focused exclusively on engineering species, which, to the reader’s surprise, it turns out, is only half the story. Furthermore, the research questions are not clearly and specifically stated. Instead, there is a vague statement on lines 92-96: “We use this framework to explore the dynamics of ecosystem assembly...” followed by results summarized into Take-Home-Messages on lines 97-120. As two of the four take-home-messages, or key insights, relate to the assembly process in networks *without* ecosystem engineers, the introduction should lay a better foundation for this *and* the aims or research questions should be made more explicit and specific.
4. **The authors should address why they evaluated the realism of the network structures only for the model without engineers.** On lines 103-105, the authors state that “...the assembly of communities in the absence of engineering reproduces many features observed in empirical systems.”; this begs the question whether the assembly model with engineers also produces

realistic structures, or not. There may be valid reasons for the scope of the performance evaluation, but, without an explanation, it seems rather odd that the model is not evaluated (also) with the engineers present. Assuming there are good reasons for this, the authors should provide them and some information on whether the evaluation of network structures without engineers has any relevance for the case with engineers or not. If it does not, it becomes even more important for the authors to motivate the model evaluation with some introductory paragraph(s), as it means that the evaluation is disconnected to anything related to ecosystem engineers, which is currently the sole focus of the introductory section.

Section 2: Assembly without ecosystem engineers

5. Accidental misleading; re-write to clarify. Lines 129-130, "Following the establishment of an autotrophic base...", together with the figure text of Figure 1b "... The basal resource is the white node rooted at the bottom of the network." misleads the reader to think that the basal resource is the same as the autotrophic base. The Method section is clear on this point, but as that section is at the end of the paper, the authors should include a bit more detail, in either the main text or figure text, to avoid any misunderstanding.
6. Line 131: Specify the type of mixotroph. I assume the authors mean the mix of auto- and heterotrophy, but mixotroph could mean other "mixes" as well, e.g. photo- and chemotrophy. The latter could be the primary colonizer as easily as a pure autotroph, which would make the logic of the assembly rule in question fall apart.
7. Lines 136-138: Provide more detail in the main text on primary extinctions. I recommend inserting here the sentence on line 495-496.
8. Lines 161-165 compares the author's assembly model with the Niche model; the point of this comparison is unclear. The paragraph is about the assembly model (without engineers) replicating empirically observed patterns of network structure. It is unclear how the comparison with the niche model fits in here. The authors should either remove this comparison or expand on it to convey what the point of the comparison is.
9. Contradictory definitions of generality; change terminology to avoid this. Line 141 read "...trophic generality (number of prey)..." while lines 169-171 defines trophic generality as in-degree scaled by link density. Note that in the latter case, in-degree is the same as generality as defined in the former case; it becomes somewhat confusing.
10. **I am not convinced that the decay in connectance is inevitable**, lines 158-159. Can the authors support that statement with a reference or self-procured evidence?

11. **Is it possible that the decreasing generality, as assembly progresses, is due to the same statistical inevitability as the decay in connectance** (assuming the authors can demonstrate this inevitability), lines 175-176?
12. Lines 183-184: “The role of specialists early on in assembly is primarily due to the accumulation of autotrophic specialists.” Are the autotrophic specialists strict herbivores or the autotrophs specialized on the basal resource? If the latter, is not then their ‘specialism’ an artifact of the single basal resource? In reality, autotrophs need multiple abiotic resources. **The authors should at least acknowledge that the choice of not including multiple basal resources makes the autotrophs artificially specialized, which in turn influences the results for generality** and how it changes with time as the assembly progresses. If this was actually the point the authors wished to make, then it needs to be made more explicitly.
13. **Specify the model setup used to produce the results**, as this is currently not clear (it is unclear in all result sections, not just this one). In the Methods or the Appendix, there should be a detailed description of the different model setups that were simulated, and which of those setups generated which results. There should also be information on this in the main text so that the reader understands the main differences in model setups that were used to produce the results in different sections. In this particular section, for example, it is completely unknown what frequencies of trophic and mutualistic interactions are used (for defining the “pool network”).

Section 3: Structure and dynamics of mutualisms

14. Rephrase lines 231-233: “Increasing service dependencies (*need* interactions; see Fig. 1) promotes both service-resource and service-service dependencies.”. It manages the feat to read both tautological and contradictory. It is not my intention to be mean here; it is just a very confusing sentence. I suggest “Increasing the number of *need* interactions (Fig. 1) increases the number of both service-resource and service-service dependencies.”.
15. Rephrase lines 238-239 “While mutualisms convey fitness advantages in order to evolve,...”. Do the authors mean that unless mutualism provided fitness advantages, evolution would not have produced this type of interaction? Right now, the sentence reads more like mutualism purposely conveys fitness advantages so that it can continue to evolve. As the authors obviously cannot mean that, one stumbles over the sentence.
16. **Clarify the design** on lines 244-246; **when increasing the frequency of mutualistic interactions, is the design substitutive or additive?** In other words, is the overall connectance held constant, and trophic interactions replaced by service interactions, or is the frequency of trophic interactions held constant, and mutualisms added, in effect increasing the connectance of the overall network (or pool)? The wording on line 371, “replace” would indicate that the design is substitutive, but I am not sure how indicative this really is.

17. **The authors need to better support the claim that “nestedness ... provides structural robustness.”** on lines 247-248. I am unconvinced by the trophic motif example. Firstly, it is unclear whether the behavior of the motif would translate to the network level. Secondly, I am unconvinced by the proposed mechanism, even for the motif. For an interaction structure to be nested, species with a lower degree need to interact with subsets of the species with which higher degree species interact. In the motif example, the high degree prey is lost from the trophic motif and the low degree interaction partner from the mutualistic motif. In both cases, nestedness is lost because there is no longer any interaction subsets. Losing nestedness this way is trivial at the small scale of the motif, but less so at the network scale. The authors should have the data to analyze whether the extinctions conform to their conceptual trophic motif example and, if they do, whether this means that the nestedness of the network is affected less by extinctions, the more mutualism there are in the network.

18. **Revise the use of the term “robustness” throughout the manuscript.** The term robustness here seems to be used synonymously with stability. This usage is perfectly valid in everyday English, but it is not a good choice in the context of the manuscript. When in the context of ecological stability, robustness has a specific meaning, namely the amount of disturbance a system (e.g. network) can withstand before a certain level of change has been reached; the R50 metric of robustness in extinction cascade studies is an example of this. Even though these types of studies have moved on to use other metrics, that do not conform to this definition of robustness, the term robustness has become strongly associated with secondary extinctions. As the manuscript at hand includes the number of secondary extinctions as a response variable, but use the term robustness to mean other things, it becomes quite confusing. For example, claiming that mutualism drives nestedness, which in turn “provides structural robustness.” (lines 244-248) clashes with the statement that increased mutualism means an increased number of secondary extinctions (lines 269-271). Avoiding the term robustness in contexts not related to secondary extinction would avoid the problem. For example, one could say instead that nestedness tends to stabilize or preserve itself. Finally, I wish to say that I understand that the authors want to avoid using the term stability, for it is used to death, and often in much too vague a manner, but if, in this case, they really only mean ‘stability’ and there is no more specific and appropriate term, I recommend them to go with ‘stability’ as being the least bad option.

19. **The authors need to better support the claim that “...mutualisms... increase the frequency of secondary extinctions (Fig. 3).”** There are no secondary extinction results in Figure 3, only results for persistence, and those results could be driven by either primary or secondary extinctions or both. Figure 4 could perhaps support the claim, but it would be better if there were an independent figure showing results for networks without engineers.

20. **Explain the calculation of nestedness.** Note that I am not asking the authors to explain the fundamentals of the nestedness metric NODF. Instead, the explanation requested relates to its implementation in this particular context; there are two question marks. Firstly, nestedness (NODF) can, to my knowledge, only be applied to bipartite networks; the networks here are certainly not bipartite. So, how did the authors manage to calculate a bipartite metric for a non-bipartite network? Secondly, is the presented NODF corrected for differences in species richness (S) and connectance (C) across the networks? The latter may not be necessary if there is no trend in either S or C with increasing frequency of mutualism. However, there is reason to believe that the decrease in persistence with increasing frequency of mutualisms, in turn, leads to a decrease in steady-state species richness (S^*) with increasing mutualism. This systematic change in S^* could bias the NODF metric. It has been a while since I worked with nestedness myself, so I may be wrong, but I would much appreciate if the authors could address these concerns in their response, and if appropriate, in the manuscript as well.
21. Rephrase lines 284-291; the wording creates a tautology and the line of argument is incomprehensible.

Section 4: Assembly with ecosystem engineering

22. Give more detail in the main text on how engineering interactions are introduced. The full description has to be left to the Methods and Appendix, of course, but there needs to be a sufficient description in the main text to enable the reader to understand the network structure for which results are presented. In the main text, the authors mention several terms which all relate to the frequency of engineering in the network: the parameter η , the frequency of engineering interactions, the number of engineering species, the number of modifiers per species, and engineering redundancies. It is from the main text very hard to understand whether these concepts are synonymous, or how they otherwise relate to each other (one has to go to the Appendix to understand this; the description in the methods is exceedingly brief and, to make matters worse, incorrect). This uncertainty, in turn, makes it hard to take in and interpret the results, which is unfortunate as this section is the main focus and novelty of the manuscript. The authors have attempted to describe this on lines 309-325, but it has to be more explicit.
23. Rewrite lines 334-355. At first read, they make a completely non-sensical impression. The problem is that the authors first state that the “nonlinear effect of engineering on rates of primary extinction results from two competing forces.” and that you then, instead of succinctly listing these two opposing forces, go into the explanation of the multi-step process of the first “force”. The impression is that the first force is described on lines 338-340 (which includes decreased secondary extinctions, which in turn makes the impression that the authors are claiming that the decrease in secondary extinctions *constitutes a part of* the decrease in primary extinctions!), and that the second force is described on lines 341-

342. It took a few reads before the paragraph made sense. It could easily be fixed by moving sentences around, such that 344-347 comes before the explanation of its underlying mechanisms, and the same with lines 353-355. Also, the use of “The first force is...”, “The second force is...”, “The mechanisms causing this force/pattern/whatever are...”, and similar explicit reader guidance would help a lot.

24. **Provide evidence for lines 344-346 “The cumulative effect in these species-rich/modifier-poor systems is increased competitive exclusion of prey... (Fig 4A)”**. There is nothing in figure 4, or elsewhere, to support the suggested mechanisms that a) it is ‘prey’ that go extinct b) due to lower vulnerability c) caused by less secondary extinctions of ‘consumers’. There is no analysis on extinction rates of species in different roles, nor anything showing that vulnerability increases with engineering frequency (or engineering presence/absence). Without such evidence the suggested mechanism is speculation, but the authors do not present it as such.
25. Similar to 24, **provide evidence for lines 353-355, specifically that competitive overlap decreases with more engineering** (higher η). Alternatively, to be more specific to the primary extinction mechanism, show that there are fewer species that meet the “at risk of primary extinction” criteria of not being the strongest competitor for any of its resources.
26. Rephrase lines 385-386 “...there exists a positive feedback between the effects of engineers on their fitness.”. It is not clear what the feedback is, and between which entities the feedback occurs.
27. Rephrase lines 426-428 “Communities lacking redundancy have lower species richness because sparse interdependencies preclude colonization”. It is not the dependencies that are sparse; if they were, this sparsity should facilitate colonization. Rather, the problem must be that the fulfillment of the dependency requirement is rare, i.e. it is often the case that no trophic and not all service needs are met.
28. **The authors should provide evidence that the result on lines 428-434 is not an artifact**. Figure S7 shows the results on colonization as a function of frequency of engineering (or, more strictly, higher η) with and without redundancies. It shows that the proportion of species in the pool that ever colonizes the local community can be as low as 0.1 (when modifiers are unique, and service frequency and η are high). Though qualitatively expected, it is quantitatively surprising and suggests that the result may be an artifact caused by cyclical dependencies. For example, if species A needs a service provided by species B, which needs species C, which needs species A, neither one of these species could ever colonize the local community, as all service needs must be met for colonization to be possible. Cyclical dependencies impeding colonization would be an issue with or without redundancy, but it should be much aggravated when there are no redundancies. It could thus cause the sharp

contrast observed in colonization rates between the case with and without redundancy. Hopefully, the results *are not* caused by this artifact, but the authors should analyze their pool networks to prove this. However, if the results are indeed driven by this artifact, I recommend that the authors change the algorithm for creating the pool networks so that such cyclical dependencies are not allowed and that they re-do the simulations, especially for the with/without redundancy contrast. Alternatively, they could keep the current simulation results and quantify to what extent these cycles contribute to the observed results.

29. Clarify lines 434-435 "...redundancy increases the niche space available to species...". I understand that engineering *per se* increases niche space, as the production of modifiers increases the potential number of resources (for food and services). But I do not understand how engineering redundancy increases niche space.
30. Improve the final paragraph, lines 441-452. Firstly, the final sentence is unclear. Are the authors saying that humans are ecosystem engineers, and thus to understand the role of engineers is to understand our own role? If so, please state that explicitly, because it would be a shame if the final sentence leaves the reader with confusion. Secondly, at this point in the manuscript, I was left feeling uncertain about what I had learned. Re-working the manuscript to make the writing more clear, and to better support the suggested mechanisms underlying the results should help with this, of course, but it would also be good if the take-home-messages (THMs) could be drummed home more decidedly. However, if Nature communications does not allow a concluding paragraph that summarizes the THMs, I recommend instead that the authors at the end of each section re-emphasize the THM(s) that pertain to that section.

Methods and Supplementary methods Appendix 1-3

31. On line 457, the authors must have meant to provide examples, rather than simply writing "(examples)".
32. On line 466, what is the "average total number of entities" an average across?
33. On line 468, does "system" relate to the pool or the local community?
34. **There are discrepancies between the method description in the Methods and the Appendix.** Here, I provide examples of the discrepancies, but it is not an exhaustive list. Instead, the authors need to go over the sections and ensure the consistency of the descriptions and the terminology. Example 1: In the methods the probability of two species eating (or needing) each other is p_e (or p_n). For species-modifier interactions the corresponding probabilities are q_e and q_n . In the methods these are termed $E_{SS}\{p_e\}$, $E_{SS}\{p_n\}$, $E_{SM}\{p_e\}$, and $E_{SM}\{p_n\}$. Example 2: In the methods, line 475 states that "each modifier is assigned to a species

independently.” Lines 476-478 then goes on to state that “This means that...there may be some modifiers that are not made by any species.”, which in light of the preceding sentence would be impossible. However, in the Appendix, lines 813-814 state that “For each species, a set number of modifiers is drawn...”, which is the reverse of what was stated on line 475 and makes lines 476-478 possible.

35. **The authors should better motivate why generalist consumers are more at risk of primary extinction than specialist ones.** Contrary to the authors’ opinion, this is *not* ecologically intuitive. I understand the argument that for any given resource, a specialist can be expected to interact (or compete) more strongly, and thus receive more benefit, than a generalist species. But I have two objections. Firstly, though this premise is appealing, and often assumed to be true, I believe there is little evidence for it. I would, however, be happy for the authors to prove me wrong on this point. Secondly, though the generalist may be the “weaker competitor” for each and all of its resources, it is likely to receive a greater grand total of “benefits” from all its interactors than the specialist. Thus, the generalist is expected to have a higher abundance than the specialist and hence be of less risk of primary extinction. This line of reasoning is in line with the observation that species extinction risk (or IUCN threat status) correlates positively with diet specialization (among other traits such as rarity, small geographic range size, etc.). I expect that the authors will be able to defend their choice sufficiently, but as this aspect of the “extinction rules” is not intuitive, their reasoning should be included in the manuscript as well as in the response to reviewers.

36. **The rules for secondary extinction are insufficiently described.** The only description is on lines 142-144: “Secondary extinctions occur when species lose its last trophic or any of its service requirements.” However, there is no mention as to how secondary extinction, and extinction cascades, are handled; are they considered events, and hence part of the event simulation process? Are secondary extinctions checked for and implemented instantaneously after a primary extinction, i.e. part of the same event? Is the same procedure followed to let longer extinction cascades play out (primary, secondary, tertiary extinctions, etc.)? The authors can put the additional description in Appendix 1.

37. **Clarify the event simulation; it is currently very opaque.** The following questions exemplify what kind of information is wanting, but there may be more missing. Thus, the authors, who know the method, should ensure they provide all necessary detail. Questions: Are there multiple events per step? If not, why is it important to define at which time within a step an event is to take place? What are the values of the event rates r_c , r_e , r_m , and r_j , and how are they calculated or chosen? How do secondary extinctions relate to these events? Is first the event type chosen and then the specific event (i.e. is it first chosen that colonization or extinction will happen, and then the species is chosen)? Or is each possible species extinction and colonization an event, and are thus the type of event and the identity of the species involved simultaneously chosen?

38. **Provide missing parameter values and explain which parameters deviate from the default** (and how) for which simulation scenario (see comment 13). For example, what are the parameter values for p_n , q_e , and q_n ?
39. Clarify why only two of the four quadrants of the pool or potential interaction matrix are relevant (line 804). It is clear why modifier-modifier interactions are irrelevant but are not modifier-species interactions as relevant as species-modifier and species-species interactions? All three of these quadrants are (partially) filled in Figure 1c.
40. **Clarify what “free parameters” signify**, line 808. Free parameters usually mean parameters that cannot be constrained by the model and instead have to be estimated. Here, I would have thought that the parameters in question were determined *a priori*, and that these were the parameters from which the pool network structure emerged. If this is not the case, the method description must be vastly improved to explain how the model works.
41. **Clarify how the equations for line 839: $E\{M_P\}_{\text{unique}}$ and Eq S3: $E\{M_P\}_{\text{redundant}}$ were arrived at.** It is not obvious, so an explanation and/or derivation is required.

Figures

42. Correct the arrow directions Figure 1a; they are contradictory. I base my interpretation of the node colors in Fig. 1a on the trophic level color legend in Fig. 1b. Assuming that this interpretation is correct, the arrows outlining the biotic interaction conform to food web conventions: in a biotic trophic interaction, the arrow shows the direction of mass and energy flow, i.e. it points from prey to predator. In a biotic service interaction, the arrow can similarly be interpreted as showing a service flow, pointing from the service provider to the service receiver. For the abiotic interactions, however, the trophic interaction arrow points from a species to a modifier, indicating mass and energy are flowing from the species to the modifier, i.e. the modifier eats the species. Similarly, for the service interaction, the arrow points from species to modifier, indicating that the species provides the modifier with a service, i.e. the modifier depends on the service from the species. In both the trophic and service species-modifier interaction, the authors must have intended for the opposite direction, i.e. the species eats or needs the modifier. For the abiotic interactions, it is only in the case of the engineering interaction that the direction of the arrow makes sense, as the arrow can be interpreted as a mass or energy flow that is required for the modifier to come into existence, i.e. be made.
43. Figure 1c is too small. It is so small that it is hard to discern the colors, especially the difference between blue and green, which makes this subplot pointless. I strongly recommend using a smaller network (i.e. fewer nodes) for illustration purposes.

44. **In figures 2a and S1a, there are discrepancies between figures and texts that need to be addressed.** In the figure texts of both figures, the steady-state species richness is reported to be reached around time step 250. However, in Figure 2a the steady state seems to be reached much earlier, around time step 10(!), and the x-axis furthermore does not even go as far as time step 250, making it a rather bad choice of figure to support the claim. Furthermore, in Figure S1a, the steady-state species richness does not seem to be reached until time step 500. First of all, there should be no discrepancy between a figure and its figure text. Secondly, there should be no discrepancy between the figures, unless there is a reason for it, e.g. they were based on simulations with different parameter configurations; if so, the authors should explicitly state this.
45. **In figures 2 and 3, explain what the measures of spread are.** There are graphic representations of variability or spread in these figures, but there is no explanation of what they are: standard errors, standard deviations, etc.
46. **In figures 2, 3, and 4, explain what the replicates are.** The figure texts mention that the results are based on a certain number of replicates. However, there is no mention of replicates in the method description, and it is not completely clear how these replicates differ.
47. In figures 3 and 4, is “Freq. mutualisms” and “Frequency of service interactions” the same thing? Do they correspond to $E_{SS}\{p_n\}$, which I take to be the frequency of service interactions in the pool community, or is it the realized frequency in the assembled community at steady-state?
48. **In figure 4, it is unclear how the response variables were calculated.** In addition to the name of the response variable, the only information on the topic is on line 332-333, which unfortunately is more confusing than clarifying. The authors can include the requested explanation in the Appendix.
49. **In figure 4d, I recommend that the authors invert the response variable,** as it would make the interpretation of the figure more intuitive. If the response variable were S^*/S^*_u , then a higher value would mean higher species richness for the ‘default’ case in which engineering redundancy is allowed. Given the context, this would be a more straightforward and intuitive response variable than S^*_u/S^* , which means that a lower value means higher species richness for the case with engineering redundancies. If the authors choose to follow this recommendation, then the same change must be made to the supplementary figure S6 as well.
50. In figure S1, it would be better if the time scale in S1a and S1b were the same. Also, clarify what is meant by “...connectance... is high because few species are tightly connected.”; what does “tightly” mean in this context?

Please find below our point-by-point responses to Reviewer comments. The reviewer comments are in **BLACK** and our responses are recorded in **BLUE**.

Sincerely,

Justin Yeakel
Mathias Pires
Marcus de Aguiar
James O'Donnell
Paulo Guimarães Jr.
Dominique Gravel
Thilo Gross

Reviewers' comments:

Reviewer #1 (Remarks to the Author):

General comment 1.0: I think this manuscript is very interesting in its approach to understand how ecosystem engineering affects networks of interacting species. The inclusion of more non-trophic interactions into such networks is extremely important if we ever want to understand the complexity of real ecological communities. Overall the manuscript reads very well and provides some interesting insights. I am in general very excited about this piece of work, but I have some questions especially about the inclusion of engineering interactions.

Response 1.0: We thank the Reviewer for their interest and appreciate their efforts in helping us address some of these loose ends. Please find our detailed responses below each comment.

Major comments:

Comment 1.1: Ecosystem engineering - as a very general description of a phenomenon where species modify their physical environment- has the benefit of making people/researchers well aware of it's importance but this concept includes a vast number of very different kind of interactions that operate at different scales. I wonder if the authors should at least spend a few sentences on that issue. For example, a tree that provides the structure for all species that live on it will lead to a massive extinction cascades if it disappears, but grazers that shorten the vegetation (e.g. rabbits) and thereby changing the soil temperature might have a less global influence. Maybe this could be addressed my changing the influence of single modifiers on different proportion of the species present? I apologise if this has been somehow included and I misunderstood.

Response 1.1: The Reviewer points to an intriguing question - how do engineers that influence different numbers of species influence community dynamics? In our model we think that the Reviewer is correct in that a modifier that many species eat or need should have a disproportionate impact than a modifier that few species eat or need. We are currently investigating how individual species and modifiers impact system stability as a function of their degree distribution and the types of interactions they are engaged in. Moreover, we do examine the influence of species-specific trophic in/out-degree on persistence, as understanding these effects provide the closest links to empirical data. Though we do not include a detailed examination of modifier in/out-degree, we do aim to examine this in a future investigation, and now elude to its potential importance in the main text:

Lines 68-71

“Engineers are widely acknowledged to have impacts on both small and large spatial scales \cite{Wright2006b}, and may serve as important keystone species in many habitats \cite{Jones2012}.”

Comment 1.2: I wasn't entirely sure how the engineering effect is included, as both positive and negative? As far as I understand this is here modelled as resources/conditions that are provided. But these can in effect impact species in very different ways. E.g. shading can allow some plant species to grow better but may slow down feeding in herbivore insects.

Response 1.2: We thank the Reviewer for this opportunity to clarify: in this current framework, there are no a priori fitness advantages attributed to ecosystem engineering (in terms of fitness for the engineer or the fitness of species using engineered resources). However the presence of an engineer in the community can positively benefit other species by providing modifiers that they eat or need. In the same vein the extinction of that engineer can negatively impact those species by eventually resulting in the loss of those modifiers, which can lead to the exclusion of those species from the community, depending on their requirements. We think that it would be of great interest to include these types of more direct negative interactions that serve to lower the fitness of some species, and may investigate this further with respect to future work. Because it would change the underlying framework of the current model, we believe that it is currently beyond the scope of this first contribution.

Comment 1.3: It would be interesting to see how ecosystem engineering affects network structure as well, does the model provide any ideas on that? Are there any structural features that might stabilize as found for other types of interactions?

Response 1.3: We thank the Reviewer for this suggestion and agree that this is a key missing piece of the analysis. In fact, this is also pointed out by Reviewer 2 (see Comment 2.4). On the one hand, it is difficult to compare a model that explicitly includes ecosystem engineering to those that do not. We now expand on this in more detail in a new Appendix 4. Importantly, we show that, when we account for the indirect links between species and engineers that produce

modifier resources, the structures of non-engineered and engineered food webs are very similar.

Lines 387-391

“While the inclusion of engineering does not significantly impact the structure of species-species interactions within assembling food webs (see Supplementary Appendix 4 and Fig. \ref{fig:trophiceng}), it does have significant consequences for community stability.”

Comment 1.4: I suspect that the estimations of secondary extinctions are very conservative as it ignores community dynamics driving secondary extinction cascades.

Response 1.4: In our model, secondary extinctions occur when a primary extinction results in a species left unable to fulfill a trophic or service dependency. In this respect, extinctions can cascade across the system due to the loss of a distantly connected species. From this perspective, secondary extinctions result directly from community dynamics (in response to previous extinctions). If we are interpreting the Reviewer’s comment correctly, this estimation of secondary extinctions may be an underestimate because we do not account for the population fluctuations and consequent stochastic extinctions that may result from community dynamics. If this is what the Reviewer means, we agree in principle but suggest that our framework is operating on a different scale. Our model is based on species’ presence/absences, where a presence implies an established population is at or near its steady state. Extinction occurs due to the loss of a species’ dependencies or due to competitive exclusion, both of which are assumed to lower its population to the point where stochastic extinction is inevitable. In lieu of modeling population dynamics directly, we model instead the changes in state that lead to conditions favorable or unfavorable to species’ occurrence in the community.

Minor comments

Comment 1.5: Abstract: please include why ecosystem engineering increases the likelihood of extinction cascades.

Response 1.5: This statement in the abstract was written wrongly (it should be the opposite). While we believe that the mechanisms proposed for these trends are likely, they are not certain, and we hesitate to emphasize them in the abstract where we do not have the space to appropriately qualify the statements.

Comment 1.6: Line 128 I guess trophic interactions are only flexible for generalists not specialists, maybe be more specific.

Response 1.6: Thank you. We have now clarified the sentence to read:

Lines 159-162:

“As such, service interactions are assumed to be obligate, whereas trophic interactions are flexible -- except in the case of a consumer with only a single resource.”

Comment 1.7: Line 141 why does increased number of prey reduce competitiveness? Is the assumption that a specialist is better adapted?

Response 1.7: Yes - we assume that generalist consumers obtain a lower profitability for each of their prey, whereas specialist consumers obtain a higher profitability for their prey. We now expand upon this concept a bit more (in the main text as well as the Supplemental Appendices) and provide references to support these assumptions.

Lines 178-181:

“This encodes three key assumptions: that mutualisms provide a fitness benefit \cite{Bronstein1994}, specialists are stronger competitors than generalists \cite{MacArthur1964,Dykhuizen1980,Futuyma1988,Costa2015}, and many predators entail an energetic cost \cite{Brown1994}.”

Comment 1.8: Line 344 This points to a negative impact of engineering on the initial set of species. But as far as I can see there are no negative effects of modifiers on species survival (e.g. this has been documented in a number of studies recording the impact of invasive ecosystem engineer species). Does that make the model too simplistic?

Response 1.8: We appreciate the Reviewer’s perspective on this and agree that there is much support for the negative effects of invasive engineers. The Reviewer is correct that we do not encode a *direct* negative effect of engineering, though the colonization of an engineer into an existing community can have *indirect* negative effects on other species. If the engineer is particularly productive such that it makes a large number of modifiers, its arrival, along with its modifiers, can facilitate colonization by species otherwise barred from the community in the absence of the engineer. The secondary arrival of these colonizers could certainly outcompete some of those present before the arrival of the engineer, such that the engineer’s invasion sets the stage for extinctions later on. This dynamic -- the immediate and downstream effects of expanding and contracting niche space due to the appearance and disappearance of species’ dependencies -- is what we set out to explore in this contribution, but we also think that including direct negative effects of engineering would be interesting to explore in future work. Such effects may be particularly rampant in microbiotic communities, where the construction of certain metabolic compounds may result in certain microbes being excluded from the community directly (via chemical incompatibility) rather than indirectly (via competitive exclusion).

Comment 1.9: Line 315 Does that mean more links from an engineer or more engineered species (links from modifier to species)?

Response 1.9: A larger number of modifiers made per species means both more engineers (species making ≥ 1 modifier) and more modifiers contributed by each engineer. These modifiers may or may not be used by the species in the community, and these links are drawn randomly when constructing the source pool (see Appendix 1). Species that eat or need a modifier have links drawn directly to the modifier rather than to the engineer. We can also consider species that eat/need modifiers to have indirect dependencies on the engineers themselves. Of course these different perspectives change our interpretation of community structure. We now consider these alternative perspectives in Appendix 4, where we evaluate the structure of engineered communities.

Reviewer 2

General Comment 2.0: This manuscript presents an ecological network assembly model, able to include feeding interactions, service interactions, and interactions mediated through ecosystem engineering. The authors analyze the model's performance, i.e. the realism of the generated network structure; they explore how structure and stability are affected by service interactions (mutualisms); and, finally, how engineering affects stability (primary and secondary extinctions; persistence; species richness at steady state).

There are four main claims. Firstly, the assembly model results in interaction networks with structures consistent with empirical observations. Secondly, increasing the frequency of mutualisms, in networks with feeding and service interactions, results in a more nested interaction structure at steady-state species richness and a lower average persistence of species in the network. Thirdly, ecosystem engineering tends to decrease primary and secondary extinctions (somewhat dependent on the frequency of engineering and service interactions). Fourthly, engineering redundancies (more than one species engineer the same entity) increases steady-state species richness through the facilitation of colonization.

To my knowledge, the main claims are novel, particularly with regards to the effects of ecosystem engineering, and are well supported by the presented data. The paper is timely and connects to the growing interest in non-trophic interactions and how they affect network structure and dynamics. The paper will certainly interest researchers in network ecology and is likely to influence thinking in the field.

For the reasons outlined above, I recommend the paper to be considered for publication in Nature communications. Due to its timeliness, novelty, and likelihood to garner interest, it is a very good fit for the journal. However, before publication, the authors need to address some issues. These issues relate more to presentation than to scientific soundness but are still vital to address (see below). I, therefore, recommend that the editor invites the authors to revise their manuscript to address these concerns before making a final decision.

Here, I briefly list the main issues (for details, see the attached full referee report):

- 1) The introduction is insufficient. Firstly, the introduction does not provide background for the first half of the study (Sections “Assembly without ecosystem engineering” and “Structure and dynamics of mutualisms”). Secondly, the authors never outline the research questions.
- 2) The method description is insufficient and opaque on several points. Furthermore, there are inconsistencies between the Methods section and Appendix 1 in describing the same aspects of the model.
- 3) There are, in some cases, inconsistencies between main text and appendix result graphs, as well as between text and result graphs.
- 4) Though the main claims are sufficiently supported, the suggestions of the underlying mechanisms are not. Some speculation into mechanism is fine, as long as the authors present it as such, but I do not find that this is the case here.

Again, I wish to emphasize that I consider these issues to be related to oversights or mistakes in the presentation, and not to faults in the science proper. As the manuscript merits further consideration, I will advise on the requested additional points:

The manuscript is in parts clearly written, in parts not. In the detailed referee comments, I have identified particular paragraphs and sentences that need revising. It is also possible that what I have mentioned as being a lack of support for the mechanisms driving the main results is simply a matter of unclear writing. Again, I have made comments on these sections, asking for clarified verbal reasoning or data to support them. I do not consider any further “experiments”, i.e. simulations, to be necessary, but a deeper analysis of existing data is, in some cases, desired (see detailed review comments).

The manuscript cannot be shortened much. If anything, the clarifications I request may require it to grow somewhat in size. However, I have identified a few paragraphs that the authors might consider for removal or shortening:

- Lines 45-56; this paper is not directly concerned with the evolutionary importance of ecosystem engineers.
- The introductory paragraphs in some of the sections seem rather broad in scope. The authors could potentially streamline them by cutting out some of the content that is not directly relevant to the paper. Paragraphs to consider for such streamlining occur on lines 210-228, 293-308, and 380-406.

The authors have done themselves justice without overselling their main claims. As far as I can tell, they are fair in their treatment of previous literature and discuss their claims appropriately in the context of previous literature, with two exceptions. Firstly, there should be some background (and hence references) that motivate the exploration of the topics of the first two result sections (i.e. model results without engineers). Secondly, the authors do not discuss the behavior of the model in relation to other assembly models, which is odd, especially given the focus of the first result section. The authors should address these two gaps in order to meet the literature-related requirements.

The authors have not provided sufficient methodological detail for the experiments to be reproduced, but the omissions are not great and should be easy to amend. In the detailed referee comments, I ask the authors to provide more methodological information on specific aspects.

There are no statistical analyses of the data, only summary statistics. Statistical analyses are not strictly necessary here. However, the descriptions of the summary statistics should be improved (see detailed comments on figures).

Response 1.0: We thank the Reviewer for their very thorough and thoughtful review. We believe that we have addressed all of the open questions and have carefully incorporated the great majority of the Reviewer's suggested edits. We think these edits and additional analyses greatly enhance the quality and clarity of our submission and thank the Reviewer for not only pointing out confusing sections, but suggesting ideas for how to fix them. As all of these comments are individually treated in more detail below, we address each one in turn. Please see our responses to each comment below.

[Detailed remarks]

Abstract

Comment 2.1: I do not agree that "ecosystem engineering...increases the magnitude of secondary extinction cascades." What is driving the increased magnitude of secondary extinction cascades are the service interactions, not the engineering interactions (see Fig. 4b). The fact that some of these service interactions are between species and modifiers, rather than between species only, does not change the causality.

Response 2.1: The Reviewer is correct in pointing out that the phrasing in the abstract is incorrect and does not match our results. We have corrected these sentences to read:

"While small numbers of engineers can increase primary extinction frequency, larger numbers of engineers both reduce primary extinction frequency as well as the size of extinction cascades."

Section 1: "Introduction"

Comment 2.2: Rephrase lines 80-82: "...service interactions account for all non-trophic interactions such as pollination or seed dispersal." All service interactions within this framework are positive for the receiver; service interactions lower the receiving species' extinction risk. Therefore, the service interactions cannot "account for all non-trophic interactions" as non-trophic interactions include negative ones, e.g. predator interference and fear of predation.

Response 2.2: We agree and have clarified the sentence to read:

Lines 110-114:

“Trophic interactions represent both predation as well as parasitism, whereas service interactions account for non-trophic interactions associated with reproductive facilitation such as pollination or seed dispersal.”

Comment 2.3: The introduction needs re-writing to better reflect what the study is about and to describe what the specific research questions are. The introductory paragraphs, lines 1-70, lead the reader to expect the study to be focused exclusively on engineering species, which, to the reader’s surprise, it turns out, is only half the story. Furthermore, the research questions are not clearly and specifically stated. Instead, there is a vague statement on lines 92-96: “We use this framework to explore the dynamics of ecosystem assembly...” followed by results summarized into Take-Home-Messages on lines 97-120. As two of the four take-home- messages, or key insights, relate to the assembly process in networks without ecosystem engineers, the introduction should lay a better foundation for this and the aims or research questions should be made more explicit and specific.

Response 2.3: We thank the Reviewer for pointing out this asymmetry. We have now balanced the introduction to include a section on prior assembly models, as well as further contextualization in the assembly model results section. Moreover, we have now pointed to key research questions that we aim to tackle prior to the Take-Home-Messages, which we think better clarifies our intent.

We now include a paragraph describing prior explorations of assembly processes in our understanding of community structure and function (see Lines 25-41). We also now begin the paragraph where we describe our aims with the two central questions that frame the goals of the manuscript:

Lines 98-102

“How does the assembly of species constrained by multitype interactions impact community structure and stability? How are these processes altered when the presence of engineers modifies species' dependencies within the community?”

Comment 2.4: The authors should address why they evaluated the realism of the network structures only for the model without engineers. On lines 103-105, the authors state that “...the assembly of communities in the absence of engineering reproduces many features observed in empirical systems.”; this begs the question whether the assembly model with engineers also produces realistic structures, or not. There may be valid reasons for the scope of the performance evaluation, but, without an explanation, it seems rather odd that the model is not evaluated (also) with the engineers present. Assuming there are good reasons for this, the authors should provide them and some information on whether the evaluation of network structures without engineers has any relevance for the case with engineers or not. If it does not, it becomes even more important for the authors to motivate the model evaluation with some

introductory paragraph(s), as it means that the evaluation is disconnected to anything related to ecosystem engineers, which is currently the sole focus of the introductory section.

Response 2.4: We thank the Reviewer for pointing this out, and note that there are some parallels between this comment and Comment/Response 1.3. Direct comparisons of our networks containing links between engineers and modifiers with those that do not account for such biotic-abiotic interactions (model or empirical) is difficult in part because the links and nodes are of a different nature. However we also acknowledge that it is of interest to assess whether and to what extent engineering impacts structure in addition to dynamics. We now include a structural analysis of engineered communities that mirrors our structural analysis of non-engineered communities, and describe this in detail in Supplemental Appendix 4.

Section 2: Assembly without ecosystem engineers

Comment 2.5: Accidental misleading; re-write to clarify. Lines 129-130, “Following the establishment of an autotrophic base...”, together with the figure text of Figure 1b “... The basal resource is the white node rooted at the bottom of the network.” misleads the reader to think that the basal resource is the same as the autotrophic base. The Method section is clear on this point, but as that section is at the end of the paper, the authors should include a bit more detail, in either the main text or figure text, to avoid any misunderstanding.

Response 2.5: We thank the Reviewer for pointing out this potentially misleading statement. While we think that the language in the text is essentially correct, we have clarified the difference between the basal resource and autotrophic base by including labels in the figure as well as the following change to the text:

Lines 162-168

“While a basal resource is always assumed to be present (white node in Fig.\ \ref{fig:model}b), following the establishment of an autotrophic base, the arrival of mixotrophs (i.e. mixing auto- and heterotrophy) and lower trophic heterotrophs create opportunities for organisms occupying higher trophic levels to invade.”

Comment 2.6: Line 131: Specify the type of mixotroph. I assume the authors mean the mix of auto- and heterotrophy, but mixotroph could mean other “mixes” as well, e.g. photo- and chemotrophy. The latter could be the primary colonizer as easily as a pure autotroph, which would make the logic of the assembly rule in question fall apart.

Response 2.6: We thank the Reviewer for catching this. We have included the parenthetical statement:

Lines 165-166

“(i.e. mixing auto- and heterotrophy)”

Comment 2.7: Lines 136-138: Provide more detail in the main text on primary extinctions. I recommend inserting here the sentence on line 495-496.

Response 2.7: We thank the Reviewer for this suggestion and have followed their advice.

8. Lines 161-165 compares the author's assembly model with the Niche model; the point of this comparison is unclear. The paragraph is about the assembly model (without engineers) replicating empirically observed patterns of network structure. It is unclear how the comparison with the niche model fits in here. The authors should either remove this comparison or expand on it to convey what the point of the comparison is.

Response 2.8: We thank the Reviewer for their comment, however we feel that the comparison of our assembly model webs with those produced from the Niche model is important for a number of reasons. First, the Niche model, though phenomenological, is widely used as a surrogate for empirical food webs due to the similarity in structure with those measured in nature. As such, showing that our assembly model webs produce structures that are -- at least in part -- similar to those produced by the niche model indicates that the assembly dynamic produces structures relevant to the real world. Second, because the niche model is well-known among both theorists and empiricists that examine food web structure, it is an important reference-point that will enable investigators in related areas to examine our results in light of prior work.

While this analysis provides a link to previous work confronting model food web structure with empirical data, it is certainly not the focus of our current contribution. As such we only briefly reference these analyses in the main text, while retaining the bulk of this discussion in the Supplement as suggested by the reviewer.

Response 2.9: Contradictory definitions of generality; change terminology to avoid this. Line 141 read "...trophic generality (number of prey)..." while lines 169-171 defines trophic generality as in-degree scaled by link density. Note that in the latter case, in-degree is the same as generality as defined in the former case; it becomes somewhat confusing.

Comment 2.9: We thank the Reviewer for pointing us to this. We have rephrased the sentence to read:

Lines 173-178

"A species' competition strength is determined by its interactions: competition strength is enhanced by the number of need interactions and penalized by the number of its resources (favoring trophic specialists) and consumers (favoring species with fewer predators)."

10. I am not convinced that the decay in connectance is inevitable, lines 158-159. Can the authors support that statement with a reference or self-procured evidence?

Response 2.10: This trend can be observed by considering the initial colonization of a small motif, which are by definition more highly connected than larger food webs. A system where connectance does not decrease would require the initial colonization of species that sparsely interact, which is not feasible in a small system if the requirement for colonization is the presence of interactions between the colonizer and those already present. As additional species colonize, it becomes more likely that those species do not interact with a large proportion of species already in the system, and connectance must decrease.

We have rephrased the sentence to read:

Lines 198-205

“Decaying connectance followed by stabilization around a constant value has been documented in the assembly of mangrove communities \cite{Piechnik2008} and experimental aquatic mesocosms \cite{Fahimipour2014}. The initial decay is likely inevitable in sparse webs as early in the assembly process the small set of tightly interacting species will have a high link density from which it will decline as the number of species increases.”

Comment 2.11: Is it possible that the decreasing generality, as assembly progresses, is due to the same statistical inevitability as the decay in connectance (assuming the authors can demonstrate this inevitability), lines 175-176?

Response 2.11: We defined a species a generalist if it consumes a proportion of the food web that is greater than or equal to the connectance (whichever measure of connectance is used). If the steady state connectance is used, a generalist in a system with few species (higher connectance) is defined relative to a larger system with many species (lower connectance). In this new version we have attempted to better match empirical measures of generality by 1) discounting autotrophic links (so that we are only assessing consumers), and 2) evaluating generalism/specialism with respect to both functional (realized) and potential trophic interactions. This has clarified the roles of assembling species, which we discuss in more detail with respect to the following Comment 2.12.

Comment 2.12: Lines 183-184: “The role of specialists early on in assembly is primarily due to the accumulation of autotrophic specialists.” Are the autotrophic specialists strict herbivores or the autotrophs specialized on the basal resource? If the latter, is not then their ‘specialism’ an artifact of the single basal resource? In reality, autotrophs need multiple abiotic resources. The authors should at least acknowledge that the choice of not including multiple basal resources makes the autotrophs artificially specialized, which in turn influences the results for generality and how it changes with time as the assembly progresses. If this was actually the point the authors wished to make, then it needs to be made more explicitly.

Response 2.12: We thank the Reviewer for pointing out these issues and agree that the prior analysis was unclear and more importantly missed the point. We have revised our measurement

of specialism/generalism in the assembling communities in two important ways. First, we have now excluded autotrophs and assessed only consumers. This is more in-line with how species trophic generality has been assessed across assembly in empirical systems and avoids some of the issues that were correctly outlined by the Reviewer above. Second, we have measured generality with respect to both functional (realized) interactions, which are plastic and change over time as the community changes, as well as potential interactions (established in the species pool interaction matrix). Functional trophic interactions are those realized at a point in time with respect to the composition of the community at that point in time. Potential trophic interactions are those determined in the species pool interaction matrix. So if a potential generalist colonizes early when only a small subset of its potential resources are available, it will functionally serve as a trophic specialist. As the reviewer anticipated, this alters (and clarifies) the interpretation of the role of trophic specialism/generality over the course of community assembly. Importantly, our findings remain consistent with prior theoretical expectations from the trophic theory of island biogeography (Gravel et al. 2011) as well as empirical observations (Piechnik et al. 2008), as we observe that the early part of the assembly process tends to favor potential generalists.

See Lines 219-228 and the revised Figure 2b.

“If generality is evaluated with respect to the steady state link density, we find that species with many potential trophic interactions realize only a subset of them, thereby functioning as specialists early in the assembly process (Fig. \ref{fig:trophic}b). As the community grows, more potential interactions become realized, and functional specialists become functional generalists. Moreover, as species assemble the available niche space expands, and the proportion of potential trophic specialists grows (Fig. \ref{fig:trophic}b).”

Comment 2.13: Specify the model setup used to produce the results, as this is currently not clear (it is unclear in all result sections, not just this one). In the Methods or the Appendix, there should be a detailed description of the different model setups that were simulated, and which of those setups generated which results. There should also be information on this in the main text so that the reader understands the main differences in model setups that were used to produce the results in different sections. In this particular section, for example, it is completely unknown what frequencies of trophic and mutualistic interactions are used (for defining the “pool network”).

Response 2.13: We thank the Reviewer for pointing out this lack of clarification on our part. While we do acknowledge the default parameterizations in the last sentence of Methods, we now include a separate Appendix (Appendix 2) that clearly articulates the different parameterizations for each section of the main text. We also refer the reader to this appendix multiple times in the main text. We believe that this will promote reproducibility as well as reader comprehension.

Section 3: Structure and dynamics of mutualisms

Comment 2.14: Rephrase lines 231-233: “Increasing service dependencies (need interactions; see Fig. 1) promotes both service-resource and service-service dependencies.” It manages the feat to read both tautological and contradictory. It is not my intention to be mean here; it is just a very confusing sentence. I suggest “Increasing the number of need interactions (Fig. 1) increases the number of both service-resource and service-service dependencies.”.

Response 2.14: We thank the Reviewer for pointing out this contradictory-tautology, and have incorporated their suggested edit.

Comment 2.15. Rephrase lines 238-239 “While mutualisms convey fitness advantages in order to evolve,...”. Do the authors mean that unless mutualism provided fitness advantages, evolution would not have produced this type of interaction? Right now, the sentence reads more like mutualism purposely conveys fitness advantages so that it can continue to evolve. As the authors obviously cannot mean that, one stumbles over the sentence.

Response 2.15: Again we thank the Reviewer for pointing out another problematic sentence. This section has been rewritten and this sentence is no longer needed.

Comment 2.16: Clarify the design on lines 244-246; when increasing the frequency of mutualistic interactions, is the design substitutive or additive? In other words, is the overall connectance held constant, and trophic interactions replaced by service interactions, or is the frequency of trophic interactions held constant, and mutualisms added, in effect increasing the connectance of the overall network (or pool)? The wording on line 371, “replace” would indicate that the design is substitutive, but I am not sure how indicative this really is.

Response 2.16: The inclusion of Appendix 2 should now clarify this, though we agree the wording in the main text can be improved. While the density of ‘eat’ interactions is not changed, what was previously a trophic interaction could become a mutualism (trophic in one direction, service in the other) if an incoming ‘eat’ interaction was paired with an outgoing ‘need’ interaction. In other words, the ‘eat’ interactions are not substituted with need interactions, though the null interaction *is* substituted with need interactions. We have clarified this with the following textual edit:

Lines 287-291

“Yet we find that as we increase the frequency of service interactions (holding constant trophic interaction frequency; see Supplementary Appendix 2), the assembled community at steady state becomes more nested (Fig. \ref{fig:nest}a).”

Comment 2.17: The authors need to better support the claim that “nestedness...provides structural robustness.” on lines 247-248. I am unconvinced by the trophic motif example. Firstly, it is unclear whether the behavior of the motif would translate to the network level. Secondly, I am unconvinced by the proposed mechanism, even for the motif. For an interaction structure to be nested, species with a lower degree need to interact with subsets of the species with which

higher degree species interact. In the motif example, the high degree prey is lost from the trophic motif and the low degree interaction partner from the mutualistic motif. In both cases, nestedness is lost because there is no longer any interaction subsets. Losing nestedness this way is trivial at the small scale of the motif, but less so at the network scale. The authors should have the data to analyze whether the extinctions conform to their conceptual trophic motif example and, if they do, whether this means that the nestedness of the network is affected less by extinctions, the more mutualism there are in the network.

Response 2.17: We thank the Reviewer for their perspective. In our example extinction probability is determined not by degree but from the competition strength, σ . In a purely trophic motif the species with the highest out-degree is more vulnerable to extinction, whereas in the mutualistic motif the species with the lowest degree (fewer received services) is more vulnerable to competitive exclusion. We now outline a more general explanation of what increases nestedness with increasing service interactions that is independent of the motif example.

Lines 283-298:

“In the absence of mutualisms, the trade-offs in our model preclude high levels of nestedness because we assume that generalists are at a competitive disadvantage when they share the same resources with a specialist consumer. Yet we find that as we increase the frequency of service interactions (holding constant trophic interaction frequency; see Appendix 2), the assembled community at steady state becomes more nested (Fig. \ref{fig:nest}a). More service interactions increase a species' competition strength, lowering its primary extinction risk. Participation in a mutualism thus delivers a fitness advantage to the species receiving the service, compensating for the lower competitive strength of generalists and allowing generalists to share subsets of resources with specialists, which promotes nestedness.”

We have now excluded description and analysis of the motif. While it provides an example of how extinctions are altered in trophic vs. mutualistic systems, it does not -- as the Reviewer indicated -- translate easily to our understanding of the broader community.

Comment 2.18: Revise the use of the term “robustness” throughout the manuscript. The term robustness here seems to be used synonymously with stability. This usage is perfectly valid in everyday English, but it is not a good choice in the context of the manuscript. When in the context of ecological stability, robustness has a specific meaning, namely the amount of disturbance a system (e.g. network) can withstand before a certain level of change has been reached; the R50 metric of robustness in extinction cascade studies is an example of this. Even though these types of studies have moved on to use other metrics, that do not conform to this definition of robustness, the term robustness has become strongly associated with secondary extinctions. As the manuscript at hand includes the number of secondary extinctions as a response variable, but use the term robustness to mean other things, it becomes quite confusing. For example, claiming that mutualism drives nestedness, which in turn “provides

structural robustness.” (lines 244-248) clashes with the statement that increased mutualism means an increased number of secondary extinctions (lines 269-271). Avoiding the term robustness in contexts not related to secondary extinction would avoid the problem. For example, one could say instead that nestedness tends to stabilize or preserve itself. Finally, I wish to say that I understand that the authors want to avoid using the term stability, for it is used to death, and often in much too vague a manner, but if, in this case, they really only mean ‘stability’ and there is no more specific and appropriate term, I recommend them to go with ‘stability’ as being the least bad option.

Response 1.18: The Reviewer is correct in detecting our difficulty in finding correct terminology for communicating community robustness, or stability. Given the documented association between robustness and secondary extinctions, we have now replaced our use of robustness as a general descriptor with ‘stability’. Where we use robustness, we do so only with respect to the prevalence of secondary extinctions.

Comment 2.19: The authors need to better support the claim that “...mutualisms... increase the frequency of secondary extinctions (Fig. 3).”. There are no secondary extinction results in Figure 3, only results for persistence, and those results could be driven by either primary or secondary extinctions or both. Figure 4 could perhaps support the claim, but it would be better if there were an independent figure showing results for networks without engineers.

Response 2.19: We agree and have reframed much of the ‘Structure and dynamics of mutualisms’ section to focus on the role of primary vs. secondary extinctions as a function of increasing frequencies of service interactions. Also see the revised Figure 3, where primary and secondary extinctions are explicitly shown and described.

Comment 2.20: Explain the calculation of nestedness. Note that I am not asking the authors to explain the fundamentals of the nestedness metric NODF. Instead, the explanation requested relates to its implementation in this particular context; there are two question marks. Firstly, nestedness (NODF) can, to my knowledge, only be applied to bipartite networks; the networks here are certainly not bipartite. So, how did the authors manage to calculate a bipartite metric for a non-bipartite network? Secondly, is the presented NODF corrected for differences in species richness (S) and connectance (C) across the networks? The latter may not be necessary if there is no trend in either S or C with increasing frequency of mutualism. However, there is reason to believe that the decrease in persistence with increasing frequency of mutualisms, in turn, leads to a decrease in steady-state species richness (S^*) with increasing mutualism. This systematic change in S^* could bias the NODF metric. It has been a while since I worked with nestedness myself, so I may be wrong, but I would much appreciate if the authors could address these concerns in their response, and if appropriate, in the manuscript as well.

Response 2.20: We have now specified that we use a unipartite measure of nestedness (UNODF), which is detailed in:

Cantor M, Pires MM, Marquitti FDM, Raimundo RLG, Sebastian-Gonzalez E, Coltri P, Perez I, Barneche D, Brandt DYC, Nunes K, Daura-Jorge FG, Floeter SR, Guimaraes PR Jr. Nestedness across biological scales. PLoS ONE 12(2): e0171691. doi:10.1371/journal.pone.0171691

The measure presented is the mean UNODF taken across columns and rows of the network, which we now specify in the main text. With respect to whether or not there is a size bias, there is a small decrease in S^* in systems with many mutualistic interactions (due to an increased risk of secondary extinctions - see Response 2.19). However, we do not find a correlation between UNODF and S^* across replicates for a given value of service interaction frequency (see Fig. S5). That the decrease in S^* is not large, and that there is no correlation with UNODF and S^* for a given service interaction frequency suggests to us that this potential bias does not strongly influence our results. We now include an additional supplemental figure showing how UNODF and S^* are uncorrelated for both low and high service interaction frequency, and include the following in the main text:

Lines 301-305

“While this shifting landscape of extinction risks lowers the steady state species richness of highly mutualistic communities, we do not observe a direct relationship between nestedness and richness (Fig. \ref{fig:nestsize}).”

Comment 2.21: Rephrase lines 284-291; the wording creates a tautology and the line of argument is incomprehensible.

Response 2.21: We thank the Reviewer for pointing this out and have rephrased the sentence to read:

Lines 337-341:

“We expect that the increased rate of secondary extinctions attributable to the loss of obligate mutualistic partners to have greater impact on system stability than the potential loss of non-obligate mutualistic partners.”

Section 4: Assembly with ecosystem engineering

Comment 2.22: Give more detail in the main text on how engineering interactions are introduced. The full description has to be left to the Methods and Appendix, of course, but there needs to be a sufficient description in the main text to enable the reader to understand the network structure for which results are presented. In the main text, the authors mention several terms which all relate to the frequency of engineering in the network: the parameter η , the frequency of engineering interactions, the number of engineering species, the number of modifiers per species, and engineering redundancies. It is from the main text very hard to understand whether these concepts are synonymous, or how they otherwise relate to each other (one has to go to the Appendix to understand this; the description in the methods is

exceedingly brief and, to make matters worse, incorrect). This uncertainty, in turn, makes it hard to take in and interpret the results, which is unfortunate as this section is the main focus and novelty of the manuscript. The authors have attempted to describe this on lines 309-325, but it has to be more explicit.

Response 2.22: We agree that it is indeed helpful to include additional detail in the textual description of engineers apart from Appendix 1 without overloading the reader with details. To this end, and hopefully in balancing these aims, we have included the following modified text:

Lines 370-386:

“The parameter η defines the mean number of modifiers produced per species in the pool, drawn from a Poisson distribution (see Methods and Supplementary Appendix 1 for details). If a species makes ≥ 1 modifier, we label it an engineer. As the mean number of modifiers/species η increases, both the number of engineers in the pool as well as the number of modifiers made per engineer increases. As detailed in Supplementary Appendix 1, multiple engineers can make the same modifier, such that engineering redundancies are introduced when η is large. When an engineer colonizes the community, so do its modifiers, which other species in the system may interact with. When engineers are lost, their modifiers will also be lost, though can linger in the community for a period of time inversely proportional to the density of disconnected modifiers in the community.”

23. Rewrite lines 334-355. At first read, they make a completely non-sensical impression. The problem is that the authors first state that the “nonlinear effect of engineering on rates of primary extinction results from two competing forces.” and that you then, instead of succinctly listing these two opposing forces, go into the explanation of the multi-step process of the first “force”. The impression is that the first force is described on lines 338- 340 (which includes decreased secondary extinctions, which in turn makes the impression that the authors are claiming that the decrease in secondary extinctions constitutes a part of the decrease in primary extinctions!), and that the second force is described on lines 341-342. It took a few reads before the paragraph made sense. It could easily be fixed by moving sentences around, such that 344-347 comes before the explanation of its underlying mechanisms, and the same with lines 353-355. Also, the use of “The first force is...”, “The second force is...”, “The mechanisms causing this force/pattern/whatever are...”, and similar explicit reader guidance would help a lot.

Response 2.23: We truly appreciate the Reviewer’s attention to the logical ordering of prose, and thank them for not only pointing out where our communication is problematic but suggesting solutions (above and beyond the expectation). We have reworded the paragraph to read:

Lines 399-418:

“When engineered modifiers are rare ($0 < \eta \leq 0.5$), higher rates of primary extinction coupled with lower rates of secondary extinction mean that extinctions are common, but of limited magnitude such that disturbances are compartmentalized. As modifiers become more common both primary and secondary extinction rates decline, which corresponds to increased

persistence. We suggest two mechanisms that may produce the observed results. First, when engineers and modifiers are present but rare, they provide additional resources for consumers. This stabilization of consumers ultimately results in increased vulnerability of prey, such that the cumulative effect is increased competitive exclusion of prey and higher rates of primary extinction (Fig. \ref{fig:engineers}a). Second, when engineers and their modifiers are common ($\eta > 0.5$) the available niche space expands, lowering competitive overlap and suppressing both primary and secondary extinctions. Notably the presence of even a small number of engineers serves to limit the magnitude of secondary extinction cascades.”

Comment 2.24: Provide evidence for lines 344-346 “The cumulative effect in these species-rich/modifier-poor systems is increased competitive exclusion of prey... (Fig 4A)”. There is nothing in figure 4, or elsewhere, to support the suggested mechanisms that a) it is ‘prey’ that go extinct b) due to lower vulnerability c) caused by less secondary extinctions of ‘consumers’. There is no analysis on extinction rates of species in different roles, nor anything showing that vulnerability increases with engineering frequency (or engineering presence/absence). Without such evidence the suggested mechanism is speculation, but the authors do not present it as such.

Response 2.24: While we emphasize that this is a ‘proposed’ mechanism, we now present a Figure S8 that we believe supports this basic idea. If we assess species-persistence as a function of trophic in-degree (the number of prey a species has) and trophic out-degree (the number of predators a species has), we observe the following support for the proposed mechanism, here described in two parts:

Mechanism Part 1: When engineers are rare, the few modifiers present support consumers to the detriment of prey, which increases primary extinction rates.

We observe that, when $\eta=0.5$, a) consumers with more prey have higher persistence in the presence of modifiers than without (which will increase predation links within the web), and b) this increased predation pressure is observed by more species exhibiting higher trophic out-degree (number of predators). Those with higher out-degree have lower persistence, though this is not observably different than for non-engineered food webs. We suggest that the correlation with increased predatory loads and higher rates of primary extinction support this part of the proposed mechanism.

Mechanism Part 2: As engineering becomes common, niche space expands, and as the presence of many modifiers in the system makes it increasingly unlikely that any two consumers share the same resources at any point in time, competitive pressure is lowered and the rates of primary extinction decrease.

We observe that, when $\eta=2.0$, there is less dependence on trophic in/out-degree with species persistence, which supports the notion that the effects of competition are more diffuse. Because competition is the sole means by which primary extinctions occur, this leads to lower rates of primary extinction.

We now point to these findings as lending support to the proposed mechanism in Figure S8, which we detail in the caption.

Lines 418-422:

“Assessment of species persistence as a function of trophic in-degree (number of resources) and out-degree (number of consumers) generally supports this proposed dynamic (Fig. \ref{fig:indeng})”

Comment 2.25: Similar to [Comment 2.]24, provide evidence for lines 353-355, specifically that competitive overlap decreases with more engineering (higher η). Alternatively, to be more specific to the primary extinction mechanism, show that there are fewer species that meet the “at risk of primary extinction” criteria of not being the strongest competitor for any of its resources.

Response 2.25: We address this in Response 2.24.

Comment 2.26: Rephrase lines 385-386 “...there exists a positive feedback between the effects of engineers on their fitness.”. It is not clear what the feedback is, and between which entities the feedback occurs.

Response 2.26: We thank the Reviewer for pointing this out and have clarified the sentence to read:

Lines 445-451:

“We argue that redundancy may be an important component of highly engineered systems, and particularly relevant when the effects of engineers increase their own fitness \cite{Cuddington2004} as is generally assumed to be the case with niche construction \cite{Krakauer2009}.”

Commnet 2.27: Rephrase lines 426-428 “Communities lacking redundancy have lower species richness because sparse interdependencies preclude colonization”. It is not the dependencies that are sparse; if they were, this sparsity should facilitate colonization. Rather, the problem must be that the fulfillment of the dependency requirement is rare, i.e. it is often the case that no trophic and not all service needs are met.

Response 2.27: We agree and have clarified the sentence to read:

Lines 495-498:

“Communities lacking redundant engineering have lower species richness because species' trophic and service dependencies are unlikely to be fulfilled, precluding colonization (Fig. \ref{fig:steadystate}c,d).”

Comment 2.28: The authors should provide evidence that the result on lines 428-434 is not an artifact. Figure S7 shows the results on colonization as a function of frequency of engineering (or, more strictly, higher η) with and without redundancies. It shows that the proportion of species in the pool that ever colonizes the local community can be as low as 0.1 (when modifiers are unique, and service frequency and η are high). Though qualitatively expected, it is quantitatively surprising and suggests that the result may be an artifact caused by cyclical dependencies. For example, if species A needs a service provided by species B, which needs species C, which needs species A, neither one of these species could ever colonize the local community, as all service needs must be met for colonization to be possible. Cyclical dependencies impeding colonization would be an issue with or without redundancy, but it should be much aggravated when there are no redundancies. It could thus cause the sharp contrast observed in colonization rates between the case with and without redundancy. Hopefully, the results are not caused by this artifact, but the authors should analyze their pool networks to prove this. However, if the results are indeed driven by this artifact, I recommend that the authors change the algorithm for creating the pool networks so that such cyclical dependencies are not allowed and that they re-do the simulations, especially for the with/without redundancy contrast. Alternatively, they could keep the current simulation results and quantify to what extent these cycles contribute to the observed results.

Response 2.28: Cyclical dependencies are not impossible but should be rare, and would certainly result in those species in the cycle being unable to colonize. They also might be expected to occur in the source pool, which is assumed to have formed via both ecological and evolutionary drivers (the latter of course is not considered in this submission). A cycle would form in exactly the way the Reviewer describes, where (A_nB, B_nC, C_nA), precluding A,B,C from the community and lowering the proportion of colonizing species. However such a cycle is expected to be *less common* if interactions are spread across more modifiers, as would be the case if each modifier made by an engineer is uniquely made. More generally, cycles will be less common in any system with more nodes (species + modifiers) in the pool. Because the number of modifiers is much greater in the unique-modifier model for a given value of η , the likelihood of forming a need-cycle is lower rather than higher. Secondly, one would expect the presence of these cycles to be greater among species with no engineers, as there are fewer entities across which interactions are spread. In other words, if cycles led to the inability to colonize, we would expect it to be most severe when $\eta=0$ and when the frequency of service interactions is high, because the nodes (species+modifiers) in the pool are fewer and the need link frequency is high. However there is not a large difference in the proportion of species capable of colonizing across the frequency of service interactions when $\eta=0$ (Figure S10c).

Comment 2.29: Clarify lines 434-435 "...redundancy increases the niche space available to species...". I understand that engineering per se increases niche space, as the production of modifiers increases the potential number of resources (for food and services). But I do not understand how engineering redundancy increases niche space.

Response 2.29: We agree that this was poorly worded. In fact, redundancy itself does not increase niche space as the Reviewer points out, but does increase the likelihood that a species' dependencies are fulfilled at a given point in time. This occurs because the extinction of a single engineer does not necessitate that its modifier immediately follow if it is made by a redundant engineer. We have clarified the sentence to read:

Lines 504-507:

"In contrast, redundancy increases the temporal stability of species' niches while minimizing priority effects by allowing multiple engineers to fulfill dependencies."

Comment 2.30: Improve the final paragraph, lines 441-452. Firstly, the final sentence is unclear. Are the authors saying that humans are ecosystem engineers, and thus to understand the role of engineers is to understand our own role? If so, please state that explicitly, because it would be a shame if the final sentence leaves the reader with confusion. Secondly, at this point in the manuscript, I was left feeling uncertain about what I had learned. Re-working the manuscript to make the writing more clear, and to better support the suggested mechanisms underlying the results should help with this, of course, but it would also be good if the take-home-messages (THMs) could be drummed home more decidedly. However, if Nature communications does not allow a concluding paragraph that summarizes the THMs, I recommend instead that the authors at the end of each section re-emphasize the THM(s) that pertain to that section.

Response 2.30: We thank the Reviewer for suggesting that we clarify this paragraph. We have modified it to read:

Lines 511-520:

"We have shown that simple process-based rules governing the assembly of species with multitype interactions can produce communities with realistic structures and dynamics. Moreover, the inclusion of ecosystem engineering by way of modifier nodes reveals that low levels of engineering may be expected to produce higher rates of extinction while limiting the size of extinction cascades, and that engineering redundancy -- whether it is common or rare -- serves to promote colonization and by extension diversity."

Lines 527-531:

"Given the rate and magnitude with which humans are currently engineering environments, understanding the role of ecosystem engineers is thus tantamount to understanding our own effects on the assembly of natural communities."

Methods and Supplementary methods Appendix 1-3

Comment 2.31: On line 457, the authors must have meant to provide examples, rather than simply writing "(examples)".

Response 2.31: The Reviewer is correct, but considering that we spend a lot of space in the main text discussing examples, we have now deleted this reference from the revision.

Comment 2.32: On line 466, what is the “average total number of entities” an average across?

Response 2.32: We have corrected the sentence to read:

Lines 544-549

“The model is initialized by creating S species and $M = \eta S$ modifiers, such that $N = S + M$ is the expected total number of entities and η is the expected number of modifiers per species in the system, where the expectation is taken across independent replicates”

Comment 2.33: On line 468, does “system” relate to the pool or the local community?

Response 2.33: System and community are used interchangeably throughout the text, though we have made sure that this is clear where ‘system’ is used. We have changed this to ‘community’ here.

34. There are discrepancies between the method description in the Methods and the Appendix. Here, I provide examples of the discrepancies, but it is not an exhaustive list. Instead, the authors need to go over the sections and ensure the consistency of the descriptions and the terminology. Example 1: In the methods the probability of two species eating (or needing) each other is p_e (or p_n). For species-modifier interactions the corresponding probabilities are q_e and q_n . In the methods these are termed $ESS\{p_e\}$, $ESS\{p_n\}$, $ESM\{p_e\}$, and $ESM\{p_n\}$. Example 2: In the methods, line 475 states that “each modifier is assigned to a species independently.” Lines 476-478 then goes on to state that “This means that...there may be some modifiers that are not made by any species.”, which in light of the preceding sentence would be impossible. However, in the Appendix, lines 813-814 state that “For each species, a set number of modifiers is drawn...”, which is the reverse of what was stated on line 475 and makes lines 476-478 possible.

Response 2.34: We thank the reviewer for pointing this out. We have trimmed Appendix 1 ensured that the parameters match the Methods. We have removed some features described previously that were not utilized in the manuscript (for example the ability to have different frequency distributions of eat and need interactions for species-species and species-modifier interactions, which we intend to explore more in a later submission). We believe that both sections are now consistent.

Comment 2.35: The authors should better motivate why generalist consumers are more at risk of primary extinction than specialist ones. Contrary to the authors’ opinion, this is not ecologically intuitive. I understand the argument that for any given resource, a specialist can be expected to interact (or compete) more strongly, and thus receive more benefit, than a

generalist species. But I have two objections. Firstly, though this premise is appealing, and often assumed to be true, I believe there is little evidence for it. I would, however, be happy for the authors to prove me wrong on this point. Secondly, though the generalist may be the “weaker competitor” for each and all of its resources, it is likely to receive a greater grand total of “benefits” from all its interactors than the specialist. Thus, the generalist is expected to have a higher abundance than the specialist and hence be of less risk of primary extinction. This line of reasoning is in line with the observation that species extinction risk (or IUCN threat status) correlates positively with diet specialization (among other traits such as rarity, small geographic range size, etc.). I expect that the authors will be able to defend their choice sufficiently, but as this aspect of the “extinction rules” is not intuitive, their reasoning should be included in the manuscript as well as in the response to reviewers.

Response 2.35: We thank the Reviewer for their perspective on this, though emphasize that this is one of the key assumptions of our model, and we have striven to articulate it as such. The underlying assumption is that a specialist will have adaptations that maximize that resource’s profitability to that consumer, whereas a generalist will not. As such the model aims to incorporate this tradeoff in strengths for specialists and generalists: a generalist will be more predisposed to competitive exclusion but more protected from secondary extinctions, whereas a specialist will be more protected from competitive exclusion yet more susceptible to secondary extinction (by losing its trophic dependencies). Of course this is a simplification, but we believe it is supported by much empirical work. Of course, opinions differ on the importance of this tradeoff, but it is the effects of this tradeoff that our model explores. The following references offer experimental and observational support for the generalist/specialist tradeoff.

Costa, A., Salvidio, S., Posillico, M., Matteucci, G., De Cinti, B. and Romano, A., 2015. Generalisation within specialization: inter-individual diet variation in the only specialized salamander in the world. *Scientific reports*, 5(1), pp.1-10.

Dykhuizen, D. and Davies, M., 1980. An experimental model: bacterial specialists and generalists competing in chemostats. *Ecology*, 61(5), pp.1213-1227.

Gottschal, J.C., de Vries, S. and Kuenen, J.G., 1979. Competition between the facultatively chemolithotrophic *Thiobacillus* A2, an obligately chemolithotrophic *Thiobacillus* and a heterotrophic *Spirillum* for inorganic and organic substrates. *Archives of Microbiology*, 121(3), pp.241-249.

MacArthur, R. and Levins, R., 1964. Competition, habitat selection, and character displacement in a patchy environment. *Proceedings of the National Academy of Sciences of the United States of America*, 51(6), p.1207.

Comment 2.36: The rules for secondary extinction are insufficiently described. The only description is on lines 142-144: “Secondary extinctions occur when species lose its last trophic or any of its service requirements.” However, there is no mention as to how secondary

extinction, and extinction cascades, are handled; are they considered events, and hence part of the event simulation process? Are secondary extinctions checked for and implemented instantaneously after a primary extinction, i.e. part of the same event? Is the same procedure followed to let longer extinction cascades play out (primary, secondary, tertiary extinctions, etc.)? The authors can put the additional description in Appendix 1.

Response 2.36: We thank the reviewer for pointing out this potential source of confusion. We now include a set of paragraphs at the end of Appendix 1 that addresses these important details of the simulation process.

Comment 2.37: Clarify the event simulation; it is currently very opaque. The following questions exemplify what kind of information is wanting, but there may be more missing. Thus, the authors, who know the method, should ensure they provide all necessary detail. Questions: Are there multiple events per step? If not, why is it important to define at which time within a step an event is to take place? What are the values of the event rates r_c , r_e , r_m , and r_j , and how are they calculated or chosen? How do secondary extinctions relate to these events? Is first the event type chosen and then the specific event (i.e. is it first chosen that colonization or extinction will happen, and then the species is chosen)? Or is each possible species extinction and colonization an event, and are thus the type of event and the identity of the species involved simultaneously chosen?

Response 2.37: Please see Response 2.36, where we have addressed all of these questions.

Comment 2.38: Provide missing parameter values and explain which parameters deviate from the default (and how) for which simulation scenario (see comment 13). For example, what are the parameter values for p_n , q_e , and q_n ?

Response 2.38: Please see Appendix 2: Model Parameterizations where we address what is set and what is altered for each section of the manuscript.

Comment 2.39: Clarify why only two of the four quadrants of the pool or potential interaction matrix are relevant (line 804). It is clear why modifier-modifier interactions are irrelevant but are not modifier-species interactions as relevant as species-modifier and species-species interactions? All three of these quadrants are (partially) filled in Figure 1c.

Response 2.39: Because we do not take advantage of quadrant-specific interaction frequencies, we have simplified our description of how the interaction pool is constructed, such that we do not now refer to interaction quadrants.

Comment 2.40: Clarify what “free parameters” signify, line 808. Free parameters usually mean parameters that cannot be constrained by the model and instead have to be estimated. Here, I would have thought that the parameters in question were determined a priori, and that these

were the parameters from which the pool network structure emerged. If this is not the case, the method description must be vastly improved to explain how the model works.

Response 2.40: We no longer use this term, and believe that we have greatly clarified how the interaction pool is constructed. Moreover we have included an Appendix 2, where we clearly articulate the different parameterizations for each section of the paper.

Comment 2.41: Clarify how the equations for line 839: $E\{MP\}_{\text{unique}}$ and Eq S3: $E\{MP\}_{\text{redundant}}$ were arrived at. It is not obvious, so an explanation and/or derivation is required.

Response 2.41: We now include intermediate steps.

Figures

Comment 2.42: Correct the arrow directions Figure 1a; they are contradictory. I base my interpretation of the node colors in Fig. 1a on the trophic level color legend in Fig. 1b. Assuming that this interpretation is correct, the arrows outlining the biotic interaction conform to food web conventions: in a biotic trophic interaction, the arrow shows the direction of mass and energy flow, i.e. it points from prey to predator. In a biotic service interaction, the arrow can similarly be interpreted as showing a service flow, pointing from the service provider to the service receiver. For the abiotic interactions, however, the trophic interaction arrow points from a species to a modifier, indicating mass and energy are flowing from the species to the modifier, i.e. the modifier eats the species. Similarly, for the service interaction, the arrow points from species to modifier, indicating that the species provides the modifier with a service, i.e. the modifier depends on the service from the species. In both the trophic and service species-modifier interaction, the authors must have intended for the opposite direction, i.e. the species eats or needs the modifier. For the abiotic interactions, it is only in the case of the engineering interaction that the direction of the arrow makes sense, as the arrow can be interpreted as a mass or energy flow that is required for the modifier to come into existence, i.e. be made.

Response 2.42: We agree whole-heartedly. Thank you for catching this mistake!

Comment 2.43: Figure 1c is too small. It is so small that it is hard to discern the colors, especially the difference between blue and green, which makes this subplot pointless. I strongly recommend using a smaller network (i.e. fewer nodes) for illustration purposes.

Response 2.43: We agree and now show a food web earlier in assembly that has fewer species and is easier to see.

Comment 2.44: In figures 2a and S1a, there are discrepancies between figures and texts that need to be addressed. In the figure texts of both figures, the steady-state species richness is reported to

be reached around time step 250. However, in Figure 2a the steady state seems to be reached much earlier, around time step 10(!), and the x-axis furthermore does not even go as far as time step 250, making it a rather bad choice of figure to support the claim. Furthermore, in Figure S1a, the steady-state species richness does not seem to be reached until time step 500. First of all, there should be no discrepancy between a figure and its figure text. Secondly, there should be no discrepancy between the figures, unless there is a reason for it, e.g. they were based on simulations with different parameter configurations; if so, the authors should explicitly state this.

Response 2.44: The time-steps have been fixed. We now use the simulation time-step consistently.

Comment 2.45: In figures 2 and 3, explain what the measures of spread are. There are graphic representations of variability or spread in these figures, but there is no explanation of what they are: standard errors, standard deviations, etc.

Response 2.45: We now plot the means (larger points), and show individual replicate results as the smaller jittered points. We clarify this in the figure captions.

Comment 4.46: In figures 2, 3, and 4, explain what the replicates are. The figure texts mention that the results are based on a certain number of replicates. However, there is no mention of replicates in the method description, and it is not completely clear how these replicates differ.

Response 2.46: Replicates are defined as the independent assembly of independently drawn source pools with a given parameterization.

Comment 4.47: In figures 3 and 4, is “Freq. mutualisms” and “Frequency of service interactions” the same thing? Do they correspond to ESS_{pn} , which I take to be the frequency of service interactions in the pool community, or is it the realized frequency in the assembled community at steady-state?

Response 4.47: We have unified the notation used in the Methods and no longer refer to these quadrant-specific interaction frequencies.

Comment 2.48: In figure 4, it is unclear how the response variables were calculated. In addition to the name of the response variable, the only information on the topic is on line 332-333, which unfortunately is more confusing than clarifying. The authors can include the requested explanation in the Appendix.

Response 2.48: We now include the following in the caption for Figure 4: “Primary and secondary extinction rates were evaluated at the community level, whereas persistence was determined for each species and then averaged across the community. Each measure reports the expectation taken across 50 replicates.”

Comment 2.49: In figure 4d, I recommend that the authors invert the response variable, as it would make the interpretation of the figure more intuitive. If the response variable were S^*/S^*_u , then a higher value would mean higher species richness for the 'default' case in which engineering redundancy is allowed. Given the context, this would be a more straightforward and intuitive response variable than S^*_u/S^* , which means that a lower value means higher species richness for the case with engineering redundancies. If the authors choose to follow this recommendation, then the same change must be made to the supplementary figure S6 as well.

Response 2.49: We agree! We thank the Reviewer for this suggestion.

Comment 2.50: In figure S1, it would be better if the time scale in S1a and S1b were the same. Also, clarify what is meant by "...connectance... is high because few species are tightly connected."; what does "tightly" mean in this context?

Response 2.50: The time-scales are nearly the same - they differ slightly because on the left we are plotting trajectories at each point in time, whereas on the right we are plotting connectance at only a few timesteps. We have rephrased the caption for S1b to read:

"Trophic connectance early in assembly is high because a small number of species interact with each other such that the proportion of realized interactions (out of all possible interactions) is closer to unity."

Reviewers' Comments:

Reviewer #1:

Remarks to the Author:

The authors have done a great job in revising the manuscript. I agree that it is reasonable to keep the model simpler at this stage (inclusion of the engineering impact). It was a pleasure to read this interesting piece of work and I believe that a better understanding of ecosystem engineering at the network level is so far lacking therefore this contribution is a necessary step.

Dirk Sanders

Reviewer #2:

Remarks to the Author:

It is the second time that I review this manuscript. The authors have made a very thorough revision and written an equally thorough Response to reviewers. The main concerns in the first round of review included insufficient and/or unclear Introduction and Method description, inconsistencies in some graphs and between some sections, and unconvincing/speculative result mechanisms; the authors have addressed all these concerns to great satisfaction. After the second round of review, I have only 11 comments, which are all minor and easily addressed. The resulting revision would be very minor and do not, in my opinion, necessitate another round of peer review. I can now wholeheartedly recommend this manuscript for publication in Nature Communications.

For the convenience of the editor, I include below the manuscript summary I made in my first review report, together with an updated version of the additional points to address for manuscripts meriting further consideration. The detailed referee report (i.e. the 11 comments) is attached as a pdf.

Manuscript and review summary:

This manuscript presents an ecological network assembly model, able to include feeding interactions, service interactions, and interactions mediated through ecosystem engineering. The authors analyze the model's performance, i.e. the realism of the generated network structure; they explore how structure and stability are affected by service interactions (mutualisms); and, finally, how engineering affects stability (primary and secondary extinctions; persistence; species richness at steady state).

There are four main claims. Firstly, the assembly model results in interaction networks with structures consistent with empirical observations. Secondly, increasing the frequency of mutualisms, in networks with feeding and service interactions, results in a more nested interaction structure at steady-state species richness and a lower average persistence of species in the network. Thirdly, ecosystem engineering tends to decrease primary and secondary extinctions (somewhat dependent on the frequency of engineering and service interactions). Fourthly, engineering redundancies (more than one species engineer the same entity) increases steady-state species richness through the facilitation of colonization.

To my knowledge, the main claims are novel, particularly with regards to the effects of ecosystem engineering, and are well supported by the presented data. The paper is timely and connects to the growing interest in non-trophic interactions and how they affect network structure and dynamics. The paper will certainly interest researchers in network ecology and is likely to influence thinking in the field.

For the reasons outlined above, I recommend the paper to be considered for publication in Nature communications. Due to its timeliness, novelty, and likelihood to garner interest, it is a very good fit for the journal.

Additional points:

- The manuscript is clearly written.
- I do not consider any further "experiments", i.e. simulations, to be necessary.
- The manuscript cannot be shortened much.
- The authors have done themselves justice without overselling their main claims.
- As far as I can tell, they are fair in their treatment of previous literature and discuss their claims appropriately in the context of previous literature.
- The authors have provided sufficient methodological detail for the experiments to be reproduced.
- There are no statistical analyses of the data, only summary statistics. Statistical analyses are not strictly necessary here.
- There are no special ethical concerns.

My thanks to the authors for their thorough consideration of and kind responses to my comments.

Alva Curtsdotter

Detailed referee comments, 2nd round, for the manuscript “Diverse interactions and ecosystem engineering stabilize community assembly” by Yeakel et al.

There are a total of 11 comments, all of them minor.

Assembly without ecosystem engineering

1. Lines 162-163: Though the use of the term “basal resource” has been much clarified through the author’s revision, I suggest changing “basal resource” to “abiotic basal resource” to preclude any misunderstanding.
2. Lines 176-178: Please specify here that a species’ competition strength is penalized by the number of its **potential** resources (i.e. number of resources in the pool) and the number of **realized** consumers (i.e. number of predators in the local community). I am aware that this is stated explicitly in the Methods, but I strongly recommend including it here in the main text, as these pieces of information are critical for understanding the effects of (low levels of) engineers on extinction rates. The suggested mechanisms for these results (lines 395-422), do not make sense without this information. I had a whole comment written up about it, before something clicked in my head and I remembered this small but critical detail...
3. Lines 205-209: I suggest the authors to add one sentence to spell out to the reader what the point is of the comparison between their assembly model and the niche model. In the Response to Reviewers they state that this is to show that their assembly model produces structures relevant to the real world and to provide a reference point to studies using the niche model. I would simply like to see the author’s to spell this out in the main text to the benefit of the reader, as it is not obvious how the comparison of network structures generated by two different models fit into the preceding comparisons of the assembly model network structure and empirical network structure. (This suggestion is similar to my original comment (2.8 in the Response to Reviewers), but I want to make clear that I agree with the author’s in a) keeping the comparison in the manuscript, and b) keeping the vast majority of the comparison in the Supplementary (Appendix 3).)
4. Lines 236-238: Here the authors write “The dominance of functional specialists early in the assembly is primarily due to the initial colonization by autotrophs.” In my first review, I asked whether this did not make the result an artifact, as there is only one abiotic basal resource for the autotrophs to feed on, so they cannot be anything but functional specialists. In the Reponse to Reviewers, 2.12, the authors agreed there was an issue here, and that their analysis was missing the point. One of the changes they made to address this, was to include only consumers in the analysis of specialism/generalism. If they included only consumers, then the importance of functional specialists early in assembly cannot be driven by the autotrophs; unless for some reason herbivores are

allowed only one autotroph prey? Is this paragraph perhaps an overlooked remnant from the first version of the manuscript? Also, if the autotrophs have been excluded from this analysis, please state that in the text, for example around lines 213-219.

Figures in main text

5. In Figure 4, there is a brief explanation of how the extinction rates and persistence are calculated. However, these metrics now occur already in Figure 3. I suggest to move or add this description to the figure text of Figure 3.
6. In Figure 4d, I suggest inverting the color scale (but keep the response variable as it is now). For these figures, the authors followed my suggestion to inverse the response variable for more intuitive reading. However, in doing so they seem to also have inverted the color scale, such that lighter colors indicate higher values and darker colors lower values. Unfortunately, in all the other subplots, in the same and other graphs, lighter colors indicate lower values and dark colors higher values. This singular deviation in the otherwise consistent color coding makes reading the subplot non-intuitive and opens up for misinterpretation unless the reader is careful. Now, this is obviously not a major issue, and I feel almost petty in pointing it out, but I also feel somewhat responsible for the issue as it happened as part of the authors, on my suggestion, trying to make the graph more intuitive to interpret!

Supplementary methods

7. I suggest that the authors, simply for greater transparency for the less mathematically minded, change equation S2 to $p_m = \frac{\eta S}{(S+M')^2} = \frac{\eta}{s(1+\eta-\frac{\eta}{\epsilon})^2}$, to clarify the definition of realized and potential links. If I got the initial step wrong, i.e. how the realized and potential links were defined, please correct it, of course.
8. In Appendix 2, I suggest including the Reponse to Reviewers 2.16 and 2.46. With regards to 2.16, the authors have improved the main text in this point, but the authors Response clarifies it further. I do not recommend putting this explanation in the main text, but it would be a valuable contribution to the Appendix. With regards to 2.46, it states explicitly how the replicates were created. Though in retrospect it seems obvious, it wasn't after my intitial read of the manuscript, so for transparenecy I would included it in the Appendix.

Supplementary figures

9. There are some inconsistencies in the figure text and labels in Figure S5. In the plot a-axis titles for mutualism it says "Mutualism in-degree (service receivers)" and "Mutualism out-degree (service donors)". Similarly, in the the figure text it says "mutualism in-degree

(the number of service receivers a species has) and out-degree (the number of service providers a species has)”. However, this definition of mutualism in- and out-degree clashes with the later sentence in the figure text: “As the mutualism in-degree increases, so does the number of service donors that are needed for the receiving species to remain in the community.” Furthermore, the results show that species persistence decreases with a species’ mutualism in-degree, while it is unaffected by mutualism out-degree. This would be consistent with the definition of mutualism in-degree being number of service donors a species has (i.e. the opposite of the definition in the figure labels and text); the more donors it has the greater is the risk for secondary extinction. This is exactly the argument the authors make in the main text, when referring to Figure S5. I’m guessing the authors got the definitions confused when labeling the figure and writing up the figure text, so it’s an easy fix.

10. I suspect the first sentence of S6 figure text is a mistake. It says that the pool consists of 200 non-engineering species, but it shows the result for networks with modifiers and engineers in it, so the pool must contain engineers.
11. In figure S9, subplot d has been updated with the inverse of the response variable, but the figure text has not been updated to reflect this change. According to the color bar, the response variable is S^*/S_u^* , but in the figure text the description is still for S_u^*/S^* . Also, comment 7, about the color scheme of Figure 4d, applies equally to Figure S9d.

Point-by-point responses to Reviewer comments and additional edits for NCOMMS-19-28516A by Yeakel et al.

Reviewer Comments:

Reviewer 1:

General Comment: The authors have done a great job in revising the manuscript. I agree that it is reasonable to keep the model simpler at this stage (inclusion of the engineering impact). It was a pleasure to read this interesting piece of work and I believe that a better understanding of ecosystem engineering at the network level is so far lacking therefore this contribution is a necessary step.

General Response: We thank Dr. Sanders very much for his constructive feedback regarding our manuscript.

Reviewer 2:

General Comment: It is the second time that I review this manuscript. The authors have made a very thorough revision and written an equally thorough Response to reviewers. The main concerns in the first round of review included insufficient and/or unclear Introduction and Method description, inconsistencies in some graphs and between some sections, and unconvincing/speculative result mechanisms; the authors have addressed all these concerns to great satisfaction. After the second round of review, I have only 11 comments, which are all minor and easily addressed. The resulting revision would be very minor and do not, in my opinion, necessitate another round of peer review. I can now wholeheartedly recommend this manuscript for publication in Nature Communications.

Manuscript and review summary:

This manuscript presents an ecological network assembly model, able to include feeding interactions, service interactions, and interactions mediated through ecosystem engineering. The authors analyze the model's performance, i.e. the realism of the generated network structure; they explore how structure and stability are affected by service interactions (mutualisms); and, finally, how engineering affects stability (primary and secondary extinctions; persistence; species richness at steady state).

There are four main claims. Firstly, the assembly model results in interaction networks with structures consistent with empirical observations. Secondly, increasing the frequency of mutualisms, in networks with feeding and service interactions, results in a more nested interaction structure at steady-state species richness and a lower average persistence of species in the network. Thirdly, ecosystem engineering tends to decrease primary and secondary extinctions (somewhat dependent on the frequency of engineering and service interactions). Fourthly, engineering redundancies (more than one species engineer the same entity) increases steady-state species richness through the facilitation of colonization.

To my knowledge, the main claims are novel, particularly with regards to the effects of ecosystem engineering, and are well supported by the presented data. The paper is timely and connects to the growing interest in non-trophic interactions and how they affect network structure and dynamics. The paper will certainly interest researchers in network ecology and is likely to influence thinking in the field.

For the reasons outlined above, I recommend the paper to be considered for publication in Nature communications. Due to its timeliness, novelty, and likelihood to garner interest, it is a very good fit for the journal.

Additional points:

- The manuscript is clearly written.
- I do not consider any further “experiments”, i.e. simulations, to be necessary.
- The manuscript cannot be shortened much.
- The authors have done themselves justice without overselling their main claims.
- As far as I can tell, they are fair in their treatment of previous literature and discuss their claims appropriately in the context of previous literature.
- The authors have provided sufficient methodological detail for the experiments to be reproduced.
- There are no statistical analyses of the data, only summary statistics. Statistical analyses are not strictly necessary here.
- There are no special ethical concerns.

General Response: We thank Dr. Curtsdotter very much for her incredibly helpful and detailed feedback on our manuscript. We would like to note that her efforts were above and beyond what is typically expected of a reviewer, and our manuscript has benefited immensely in both its clarity and content. Our sincere gratitude!

Specific Comments

Assembly without ecosystem engineering

Comment 2.1: Lines 162-163: Though the use of the term “basal resource” has been much clarified through the author’s revision, I suggest changing “basal resource” to “abiotic basal resource” to preclude any misunderstanding.

Response 2.1: We have incorporated the suggested edit.

Comment 2.2: Lines 176-178: Please specify here that a species’ competition strength is penalized by the number of its potential resources (i.e. number of resources in the pool) and the number of realized consumers (i.e. number of predators in the local community). I am aware that this is stated explicitly in the Methods, but I strongly recommend including it here in the main text, as these pieces of information are critical for understanding the effects of (low levels of) engineers on extinction rates. The suggested mechanisms for

these results (lines 395-422), do not make sense without this information. I had a whole comment written up about it, before something clicked in my head and I remembered this small but critical detail...

Response 2.2: Yes - thank you for pointing out this omission. We have now included this clarification. We note that for need interactions, the realized and potential interactions are the same, so we have not included that distinction with respect to services. The sentence now reads:

Line 169: *“A species' competition strength is determined by its interactions: competition strength is enhanced by the number of need interactions (where the number of potential and realized interactions are equivalent) and penalized by the number of its realized resources (i.e. those resources present in the local community, favoring functional trophic specialists) and realized predators (i.e. those predators present in the local community)..”*

Comment 2.3: Lines 205-209: I suggest the authors to add one sentence to spell out to the reader what the point is of the comparison between their assembly model and the niche model. In the Response to Reviewers they state that this is to show that their assembly model produces structures relevant to the real world and to provide a reference point to studies using the niche model. I would simply like to see the author's to spell this out in the main text to the benefit of the reader, as it is not obvious how the comparison of network structures generated by two different models fit into the preceding comparisons of the assembly model network structure and empirical network structure. (This suggestion is similar to my original comment (2.8 in the Response to Reviewers), but I want to make clear that I agree with the author's in a) keeping the comparison in the manuscript, and b) keeping the vast majority of the comparison in the Supplementary (Appendix 3).)

Response 2.3: Thank you for this feedback. We have modified this section to read:

L203: *“In Supplementary Note 3 we include a brief comparison of assembly model food webs with those produced by the Niche model \cite{Williams2000}. While the aims of these approaches are quite distinct, we provide this comparison as a reference point to traditional food web models, and to emphasize that both approaches result in food webs with similar structures (Supplementary Figures 2,3).”*

Comment 2.4: Lines 236-238: Here the authors write “The dominance of functional specialists early in

the assembly is primarily due to the initial colonization by autotrophs.” In my first review, I asked whether this did not make the result an artifact, as there is only one abiotic basal resource for the autotrophs to feed on, so they cannot be anything but functional specialists. In the Reponse to Reviewers, 2.12, the authors agreed there was an issue here, and that their analysis was missing the point. One of the changes they made to address this, was to include only consumers in the analysis of specialism/generalism. If

they included only consumers, then the importance of functional specialists early in assembly cannot be driven by the autotrophs; unless for some reason herbivores are allowed only one autotroph prey? Is this paragraph perhaps an overlooked remnant from the first version of the manuscript? Also, if the autotrophs have been excluded from this analysis, please state that in the text, for example around lines 213-219.

Response 2.4: We agree with the Reviewer and note that the description of concern was indeed left over from the previous version of the manuscript. We have amended the following sentences to read:

Line 218: *“Only trophic links between species are considered here, such that we ignore links to the abiotic basal resource in our evaluation of trophic generality.”*

Line 239: *“The dominance of functional specialists early in assembly is primarily due to the initial colonization by consumers with few resources.”*

Figures in main text

Comment 2.5: In Figure 4, there is a brief explanation of how the extinction rates and persistence are calculated. However, these metrics now occur already in Figure 3. I suggest to move or add this description to the figure text of Figure 3.

Response 2.5: We have now amended Figures 3 and 4 to be consistent in how they define primary and secondary extinction rates and persistence.

Comment 2.6: In Figure 4d, I suggest inverting the color scale (but keep the response variable as it is now). For these figures, the authors followed my suggestion to inverse the response variable for more intuitive reading. However, in doing so they seem to also have inverted the color scale, such that lighter colors indicate higher values and darker colors lower values. Unfortunately, in all the other subplots, in the same and other graphs, lighter colors indicate lower values and dark colors higher values. This singular deviation in the otherwise consistent color coding makes reading the subplot non-intuitive and opens up for misinterpretation unless the reader is careful. Now, this is obviously not a major issue, and I feel almost petty in pointing it out, but I also feel somewhat responsible for the issue as it happened as part of the authors, on my suggestion, trying to make the graph more intuitive to interpret!

Response 2.6: :) We appreciate the Reviewer’s attention to detail and have reversed the color scheme accordingly.

Supplementary Information

Comment 2.7: I suggest that the authors, simply for greater transparency for the less mathematically minded, change equation S2 to $p_m = \eta S(S+M)^2 = \eta S(1+\eta - \eta e)^2$, to clarify the definition of realized and potential links. If I got the initial step wrong, i.e. how the realized and potential links were defined, please correct it, of course.

Response 2.7: We have included the suggested intermediate calculation.

Comment 2.8: In Appendix 2, I suggest including the Reponse to Reviewers 2.16 and 2.46. With regards

to 2.16, the authors have improved the main text in this point, but the authors Response clarifies it further. I do not recommend putting this explanation in the main text, but it would be a valuable contribution to the Appendix. With regards to 2.46, it states explicitly how the replicates were created. Though in retrospect it seems obvious, it wasn't after my initial read of the manuscript, so for transparency I would included it in the Appendix.

Response 2.8: Agreed. Supplementary Note 2 now includes these clarifications.

Supplementary figures

Comment 2.9: There are some inconsistencies in the figure text and labels in Figure S5. In the plot a-axis

titles for mutualism it says "Mutualism in-degree (service receivers)" and "Mutualism out-degree (service donors)". Similarly, in the the figure text it says "mutualism in-degree (the number of service receivers a species has) and out-degree (the number of service providers a species has)". However, this definition of mutualism in- and out-degree clashes with the later sentence in the figure text: "As the mutualism in-degree increases, so does the number of service donors that are needed for the receiving species to remain in the community." Furthermore, the results show that species persistence decreases with a species' mutualism in-degree, while it is unaffected by mutualism out-degree. This would be consistent with the definition of mutualism in-degree being number of service donors a species has (i.e. the opposite of the definition in the figure labels and text); the more donors it has the greater is the risk for secondary extinction. This is exactly the argument the authors make in the main text, when referring to Figure S5. I'm guessing the authors got the defintions confused when labeling the figure and writing up the figure text, so it's an easy fix.

Response 2.9: We thank the Reviewer for catching this and have fixed the discrepancy.

Comment 2.10: I suspect the first sentence of S6 figure text is a mistake. It says that the pool consists of

200 non-engineering species, but it shows the result for networks with modifiers and engineers in it, so the pool must contain engineers.

Response 2.10: Correct - this mistake is now fixed.

Comment 2.11: In figure S9, subplot d has been updated with the inverse of the response variable, but the figure text has not been updated to reflect this change. According to the color bar, the response variable is S^*/Su^* , but in the figure text the description is still for Su^*/S^* . Also, comment 7, about the color scheme of Figure 4d, applies equally to Figure S9d.

Response 2.11: Another great catch. This has now been fixed and the color scheme has been updated.

Additional changes to the manuscript:

- 1) The title has been changed to that suggested by the editor
- 2) The abstract has been shortened to 164 words
- 3) The third paragraph of the introduction has been partly removed and partly merged with the second paragraph, as it was largely repetitive and ill-placed.
- 4) Minor changes to grammar throughout and general formatting to conform to NatComm guidelines